# Structural basis for sequence-independent substrate selection by eukaryotic wobble base tRNA deaminase ADAT2/3

Luciano G. Dolce[1], Aubree A. Zimmer[2], Laura Tengo [1], Félix Weis [3], Mary Anne T. Rubio [2], Juan D. Alfonzo [2] & Eva Kowalinski [1]✉

The essential deamination of adenosine $A_{34}$ to inosine at the wobble base is the individual tRNA modification with the greatest effects on mRNA decoding, empowering a single tRNA to translate three different codons. To date, many aspects of how eukaryotic deaminases specifically select their multiple substrates remain unclear. Here, using cryo-EM, we present the structure of a eukaryotic ADAT2/3 deaminase bound to a full-length tRNA, revealing that the enzyme distorts the anticodon loop, but in contrast to the bacterial enzymes, selects its substrate via sequence-independent contacts of eukaryote-acquired flexible or intrinsically unfolded motifs distal from the conserved catalytic core. A gating mechanism for substrate entry to the active site is identified. Our multi-step tRNA recognition model yields insights into how RNA editing by $A_{34}$ deamination evolved, shaped the genetic code, and directly impacts the eukaryotic proteome.

All nucleic acids in cells undergo post-transcriptional or post-replicative chemical modification, with tRNAs displaying the largest diversity of modified nucleotides. Modifications can affect the stability, folding and ultimately the function of tRNA in translation; in many cases RNA modification defects are associated with neurological disorders, cancers, or other diseases. In general, modifications in the core of a tRNA may sustain the canonical tRNA structure, while those occurring in the anticodon loop may fine-tune and streamline the essential process of protein synthesis at the ribosome[1–5]. Given the extraordinary degree of conservation in the 3D architecture of tRNAs, with their characteristic L-shape, it is fundamental to understand how different tRNA interacting proteins, including modification enzymes, can specifically recognize their cognate substrate tRNA and refuse structurally similar non-substrates. This is particularly critical for modifiers of the anticodon domain, as their action directly affects codon recognition and may influence the integrity of the cellular proteome[6–9].

Several modifications occur at the first position of the anticodon (position 34) of tRNAs, the so-called wobble position, which base pairs

to the third codon position of a base triplet in mRNAs[5]. Bacterial genomes do not contain any gene for a tRNA that per se can read the cytosine-ending (C-ending) codon triplet encoding arginine. Similarly, eukaryotes do not encode tRNAs that as such would read the C-ending codons for alanine, isoleucine, leucine, proline, serine, threonine, valine, and arginine[10]. To resolve this dilemma, a single tRNA (tRNA^Arg_ACG) in bacteria and 7–8 different tRNAs (depending on the organism) with an encoded $A_{34}$ in eukaryotes (i.e. tRNA^Thr_AGU, tRNA^Ala_AGC, tRNA^Pro_AGG, tRNA^Ser_AGA, tRNA^Leu_AAG, tRNA^Ile_AAU, tRNA^Val_AAC, and tRNA^Arg_ACG) are post-transcriptionally modified to inosine ($I_{34}$). This $A_{34}$ to $I_{34}$ tRNA deamination leads to the greatest enhancement in decoding capacity that can be caused by a single modification, as the INN anticodon can base pair with three different codon triplets at the mRNA, namely NNC (cytosine-ending), NNU (uracil-ending) and to a lesser extent NNA (adenosine-ending)[11–16].

$I_{34}$ formation is essential for viability and catalyzed by adenosine deaminases acting on tRNA (ADATs)[17–20]. Several RNA-free bacterial tRNA deaminase A (TadA) structures were reported[21–23], followed by the *Staphylococcus aureus* enzyme bound to a synthetic hairpin

[1]EMBL Grenoble, 71 Avenue des Martyrs, 38042 Grenoble, France. [2]Department of Microbiology and The Center for RNA Biology, The Ohio State University, Columbus, OH, USA. [3]EMBL Heidelberg, Structural and Computational Biology Unit, Meyerhofstraße 1, 69117 Heidelberg, Germany. ✉e-mail: kowalinski@embl.fr

mimicking the anticodon-stem loop of bacterial tRNA$^{Arg}$. TadA is a homodimeric enzyme, which binds one RNA ligand per protomer, forming a 2:2 protein: RNA complex. In this complex, the tRNA anticodon loop adopts an unusual conformation where the nucleosides are splayed out in a manner that is different from a canonical anticodon loop. TadA makes sequence-specific contacts to the exposed nucleobases[24]. In eukaryotes, the essential enzyme catalyzing I$_{34}$ formation in tRNA is a heterodimer, composed by one active (ADAT2) and one inactive (ADAT3) paralogous subunits, both homologs of the bacterial enzyme[11,12,18,22,25,26]. These ADATs carry a characteristic cytidine deaminase (CDA) active-site amino acid signature motif (C/H) XEX$_n$PCXXC (with X being any amino acid, and n being any number of residues), containing the active-site acidic glutamate (inert valine in inactive subunit ADAT3) and cysteine/histidine residues coordinating a Zn$^{2+}$ cation[21,27,28]. The motif, shared with other members of the CDA superfamily, is also found in the C-to-U deaminases which include the AID/APOBEC family.

In humans, mutations in the ADAT3 gene cause autosomal recessive rare disorders manifesting in a spectrum of intellectual disabilities and the tRNA pools isolated from affected individuals showed defects in A$_{34}$-to-I$_{34}$ deamination[29–35]. More recent investigations aimed to understand how the increased number of NNA codons in eukaryotic tRNAs pools and the expanded substrate recognition capacity by ADAT2/3 are related and how this could affect the proteome[8,11,36]. Furthermore, it has been reported that I$_{34}$ in tRNA impacts gene expression regulation during differentiation of pluripotent stem cells by improving translation efficiency[9].

Unlike bacterial TadA, which in vitro is also active on smaller harpin substrates, the eukaryotic ADAT2/3 requires a complete full-length tRNA L-shape for efficient deamination. Moreover, a classical 7-nucleotide anticodon loop has been shown to be a key determinant, since adding extra nucleotides to the loop interfered with I$_{34}$ formation in vitro[37,38]. All tRNAs meeting these requirements, target and non-target tRNAs, can interact with ADAT2/3, yet, in a previous study, an A$_{34}$ containing substrate displayed a ten times faster dissociation compared to other nucleotides[39]. However, the structures of the ADAT2/3 enzymes in the absence of a tRNA ligand are not sufficient to explain how the tRNA architecture is recognized by ADAT2/3[40,41].

Here we present the structure of an ADAT deaminase bound to a full-length tRNA. Our cryo-electron microscopy (cryo-EM) structure of the tRNA-bound ADAT2/3 from *Trypanosoma brucei* (*T. brucei* or *Tb*) reveals a conserved distortion of the anticodon loop and exposure of the nucleotide bases in common with the bacterial catalytic pocket, with most contacts to the anticodon loop and the target nucleotide A$_{34}$ provided by the active ADAT2 subunit. Through the evolution from a homo- to a heterodimeric configuration, both ADAT2/3 subunits have evolved additional peripheral binding regions, which provide sequence-independent interactions that help to select and correctly position the tRNA substrate for catalysis. More generally, our findings provide insights into the evolution of these essential tRNA deamination enzymes.

## Results

### The cryo-EM structure of the *Trypanosoma brucei* ADAT2/3-tRNA complex

To reveal how the eukaryotic tRNA deaminase (ADAT2/3) interacts with its substrate, we reconstituted the *T. brucei* protein-tRNA complex. Co-expression in insect cells and purification via a single 8-his-tag in *Tb*ADAT3 yielded a stoichiometric 1:1 *Tb*ADAT2-*Tb*ADAT3 complex and circumvented the enrichment of homodimers formed by the excess of *Tb*ADAT2 in the final sample, as *Tb*ADAT3 alone is insoluble (Supplementary Fig. 1a)[17]. Regardless of many attempts, the heterodimeric *Tb*ADAT2/3 complex failed to crystallize despite its extreme stability in stringent wash conditions (1 M KCl) during purification, and its elevated melting temperature of approximately 53.5 ˚C in

thermostability assays (Supplementary Fig. 1b). Co-purification of cellular tRNA$^{Thr}$ and tRNA$^{Pro}$ in initial purifications from *E. coli* (Supplementary Fig. 1c), led us to speculate that a tRNA indeed would stabilize the complex and create a particle size amenable for cryo-EM single-particle analysis. The 1:1:1 *Tb*ADAT2:*Tb*ADAT3:*Tb*tRNA$^{Thr}$$_{CGU}$ complex was reconstituted with in vitro transcribed *Tb*tRNA$^{Thr}$$_{CGU}$, a non-cognate tRNA omitting the reactive A$_{34}$ that we chose to stabilize the complex by preventing product turnover (Supplementary Fig. 1d). The trimeric 95 kDa complex behaved well on quantifoil ultrAufoil gold grids and the preferred orientation of the particles was compensated by tilting the stage 30˚ for data collection. Data was processed with a combination of available software suites, and the resolution of the final map was calculated by the gold standard fourier shell correlation method using the cutoff of 0.143 by four programs: cryoSPARC 3.62 Å, phenix 3.66 Å, PDBe FSC server 4.08 Å, and 3DFSC 4.12 Å. This resolution range is reasonable with respect to particle size (Supplementary Fig. 2 and Supplementary Table 1)[42,43]. We rigid-body fitted the *Sc*tRNA$^{Phe}$ (PDB: 1EHZ) but refined only the portion of the *Tb*tRNA$^{Thr}$$_{CGU}$ anticodon loop, which was well resolved (Supplementary Fig. 3a). The homologous yeast (PDB: 7BV5) and mouse (PDB: 7NZ7, 7NZ8) apo-crystal structures, and a model of the dimeric complex generated with AlphaFold served as templates for building and refinement of the heterodimeric *Tb*ADAT2/3 deaminase core (Supplementary Fig. 1e, f)[44]. The N-terminal domain of *Tb*ADAT3 (*Tb*ADAT3$^N$) displayed poorer density, but a DeepEMhancer post-processed map allowed jiggle-fitting the AlphaFold model, without further refinement interventions[45]. The final model comprises the tRNA molecule, *Tb*ADAT2 residues 19–103, 123–175, and 183–221, and *Tb*ADAT3 residues 1–55, 105–263, and 277–340 (Fig. 1a, c).

### The anticodon loop of the tRNA is remodeled upon ADAT2/3 binding

We inspected the overall conformation of the tRNA in the complex and noticed that the conformation of the anticodon-stem loop bound to the *Tb*ADAT2/3 core dramatically deviates from canonical free or ribosome A-site bound tRNA structures (Fig. 2a). In our complex structure, the *Tb*tRNA$^{Thr}$$_{CGU}$ is deeply inserted into the active site of *Tb*ADAT2 with all seven nucleotides (32–38) of the anticodon loop interacting directly with the ADAT2/3 deaminase core (Fig. 2b). While the stem nucleotides strictly obey Watson-Crick base pairing, and the non-canonical base transition pair between C$_{32}$ and A$_{38}$ is maintained[46], the loop nucleotides 33–37 are extensively remodeled. Here, the sugar-phosphate backbone adopts a bent conformation and the nucleosides U$_{33}$, C*$_{34}$, G$_{35}$ and A$_{37}$ are splayed outwards (Fig. 2c and Supplementary Fig. 3a). The nucleobase of U$_{36}$ is unconventionally internalized via a π-π stacking interaction between the nucleobase of C$_{32}$ and residue Y205 of ADAT2 (Fig. 2d and Supplementary Fig. 3a). The unusual ADAT2/3-bound anticodon loop conformation with exposed bases, similar to the TadA bound hairpin, was unexpected, because the eukaryotic deaminase does not rely on sequence read-out[24,37,41].

### A deep catalytic pocket in ADAT2/3 accommodates A$_{34}$

To determine the molecular basis for the unusual rearrangement of the tRNA, we examined how the binding pocket accommodates the anticodon loop. This pocket lies in the dimer interface between the cytidine deaminase (CDA) domains of *Tb*ADAT2 and *Tb*ADAT3. The flipped-out A$_{34}$ (C$_{34}$ in our construct) is deeply inserted into the active site of *Tb*ADAT2 (Fig. 2b). The active site of *Tb*ADAT2 features the typical CDA elements: a Zn$^{2+}$ cation coordinated by two cysteines (C136$^{Tb ADAT2}$ and C139$^{Tb ADAT2}$) and a histidine (H90$^{Tb ADAT2}$), and the catalytic glutamate (E92$^{Tb ADAT2}$), suggesting a similar catalytic mechanism as described for TadA[24]. Furthermore, the reactive nucleobase C*$_{34}$ conformation is stabilized through hydrophobic interactions with V46$^{Tb ADAT2}$ and V133$^{Tb ADAT2}$ as well as hydrogen bonding with the conserved asparagine N79$^{Tb ADAT2}$, while its ribose moiety is clamped by

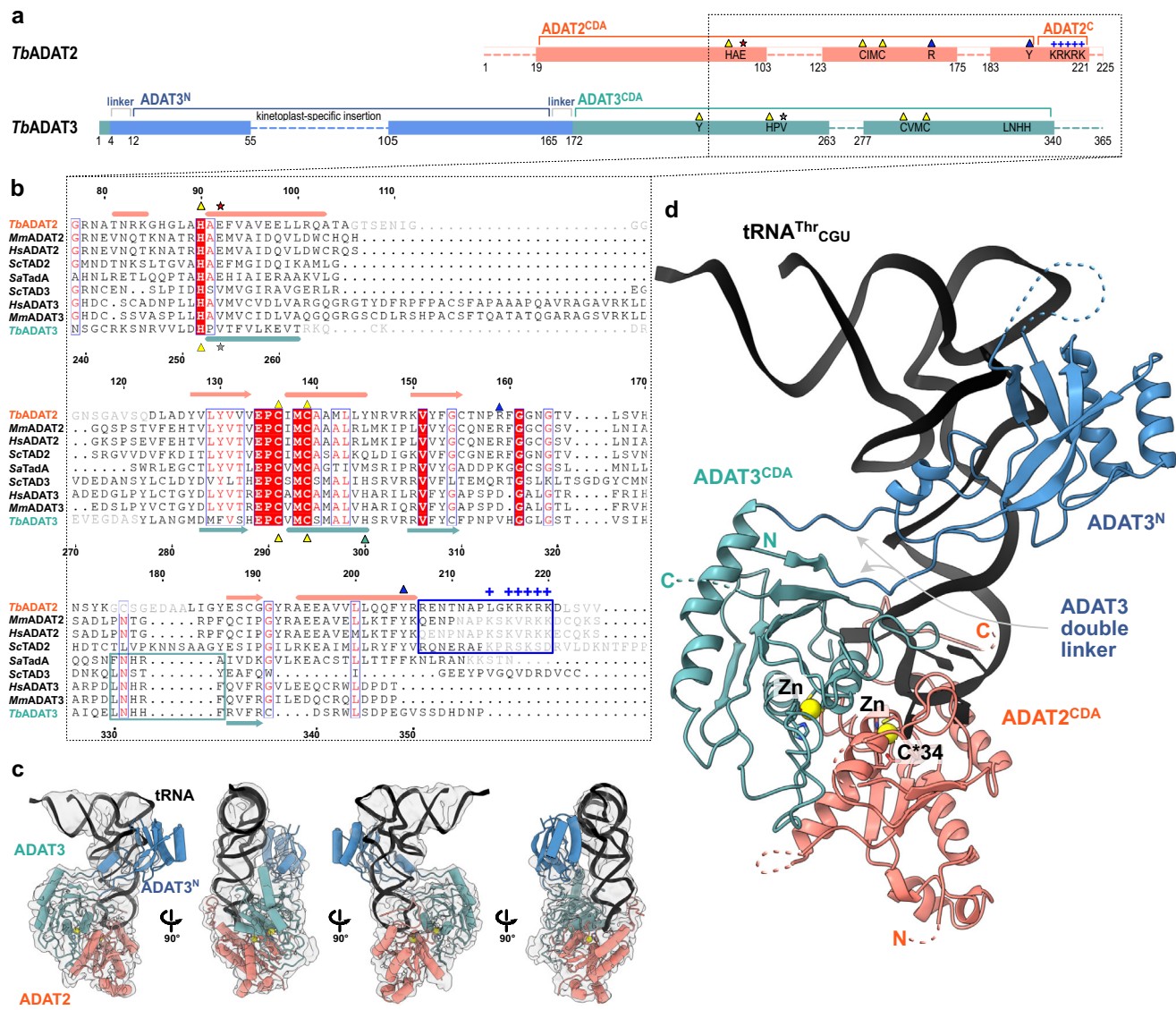

**Fig. 1 | The structure of *Tb*ADAT2/3 heterodimer bound to tRNA^Thr_CGU.**
**a** Schematic representation of the protein subunits: ADAT2 CDA in salmon, ADAT3 N-terminal domain in blue, ADAT3^CDA in teal. Portions with traced atomic model as solid colors and non-resolved linkers as dotted lines. Important sequence motifs are annotated at their relative positions. Zn^2+ coordinating residues are indicated by yellow triangles, the active-site glutamate / pseudo-active-site valine by a red and gray star, 'gate' residues of ADAT2 with a blue triangle. Conserved positive charges of the KR-motif annotated as blue "+" signs. **b** Structural alignment of *Tb*, yeast and mouse ADAT2 and ADAT3 CDA domains and *Staphylococcus aureus* TadA. The sequences for human (*Hs*) ADAT2/3 included (*Mm*ADAT2/3 PDB: 7NZ7, *sc*TAD2/3 PDB:7BV5, *sa*TadA PDB: 2B3J). Structural elements of *Tb* are annotated (ADAT2 above, ADAT3 below) with α-helices as cylinders and β-sheets as arrows. Zn^2+ coordinating residues are indicated by yellow triangles, the active-site glutamate / pseudo-active-site valine by a red and gray star, 'gate' residues of ADAT2 with a blue triangle and the active-site histidine of ADAT3 with a teal triangle. ADAT2^C is boxed in blue with annotation of the conserved positive charges of the KR-motif ("+" signs). Grayed out letters indicate non-resolved loops in our structure and ADAT2^C residues not resolved in previous crystal structures. An ADAT3 conserved loop is annotated as a teal box. Other colors and boxes within the alignment represent relative conservation above 70%. **c** Four views of the final non-b-factor-sharpened cryoSPARC map that was used for modeling (presented at a level of 0.09) with atomic model. Colors as above. **d** Cartoon model of ADAT2/3 bound to tRNA. Colors as in 1 A. Zn^2+ atoms in yellow, tRNA in black with only anticodon loop bases represented as slabs, the reactive C_34 nucleotide is annotated as C*. Gray arrows point to the ADAT3 N-terminal domain double linker.

V44^*Tb*ADAT2 and F160^*Tb*ADAT2 (Fig. 2e and Supplementary Fig. 3b). The catalytic glutamic acid side chain, E92^*Tb*ADAT2, and the catalytic water could not fully be resolved in our maps (Supplementary Fig. 3b). Besides these observations, the binding pocket for nucleotide A_34 is remarkably similar between the tRNA-bound and -unbound structures, pointing to a certain rigidity of the active site to which the tRNA ligand must adapt (compare Fig. 2e, f, and Supplementary Fig. 3c). Remarkably, most residues in direct proximity to A_34 are conserved between the bacterial and eukaryotic enzymes (Fig. 2g). An exception is the presence of histidine H300^*Tb*ADAT3, which is conserved within ADAT3s (Fig. 1b, compare Figs. 2e, g). In summary, the central ADAT2/3

catalytic pocket for the C*_34 nucleotide base is highly conserved between the bacterial homodimer and the eukaryotic heterodimer and no large conformational changes can be observed between the available eukaryotic substrate-bound and -unbound A_34 pockets. Taken together these observations imply that for binding, the tRNA must adapt to ADAT2/3 through structural rearrangements.

**The anticodon loop is anchored by a molecular gate in ADAT2**
In the bacterial enzyme TadA, extensive remodeling of the anticodon loop is provoked by favorable base-specific interactions, resulting in splayed-out anticodon loop nucleosides bound to a rather rigid

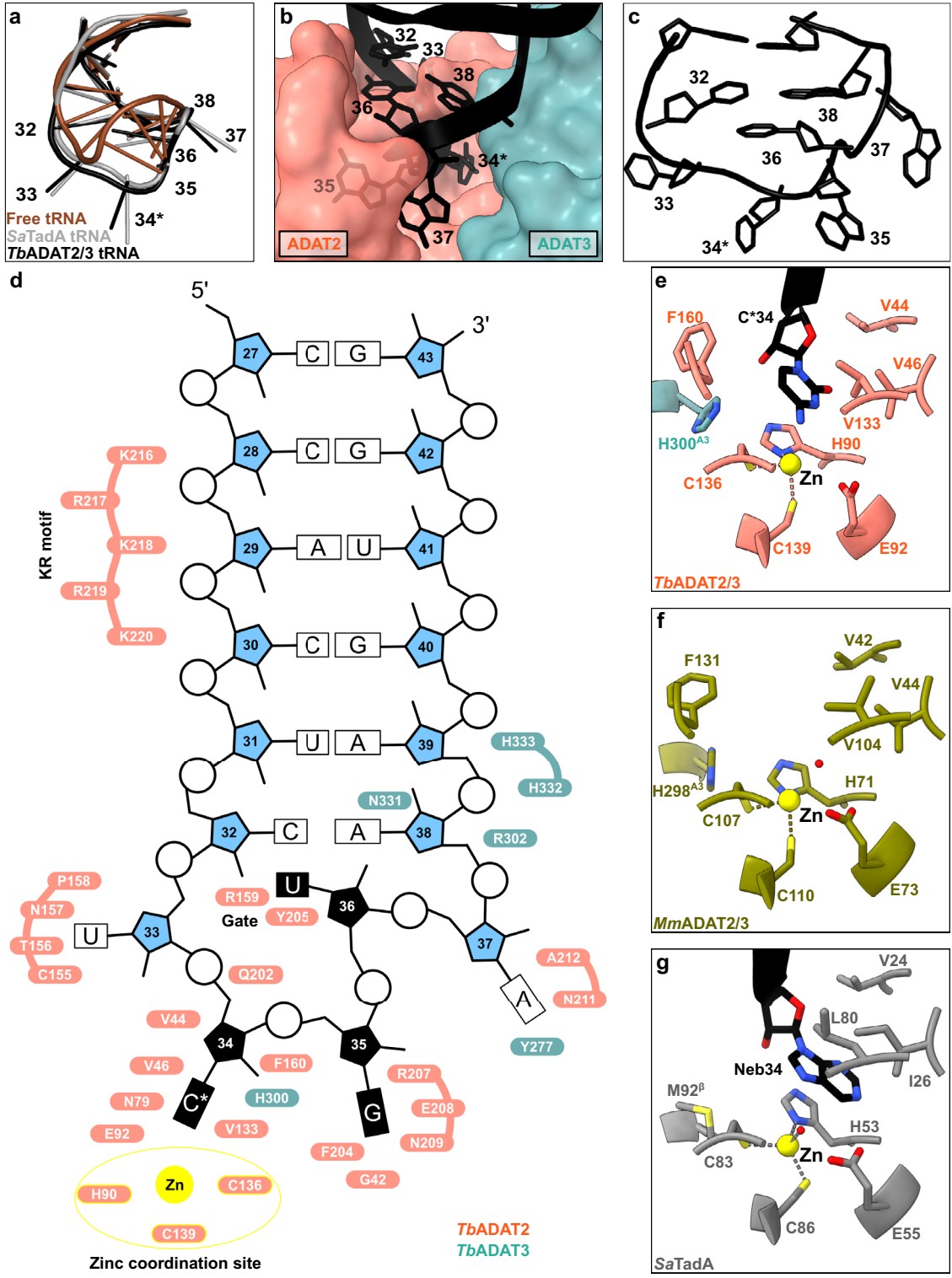

**Fig. 2 | The interaction of the tRNA anticodon-stem loop with ADAT2/3. a** The anticodon loop (ACL) bound to *Tb*ADAT2/3 (black) is distorted with respect to free tRNA (brown; PDB: 1EHZ) but resembles the ACL bound to bacterial TadA (gray; PDB: 2B3J), all cartoon-stick representation. **b** A deep cleft in the dimer interface accommodates the ACL. Colors as in Fig. 1, ADAT2/3 model as surface representation, tRNA as cartoon with anticodon loop bases shown. **c** The deformed ACL bound to the *Tb*ADAT2/3, backbone in cartoon and bases in stick representation. **d** Schematic overview of the anticodon stem loop (ACSL) and its interactions with

*Tb*ADAT2/3. Zinc coordinating site highlighted in yellow. **e**–**g** Comparison of the active-site pockets of *Tb* and *Mm*ADAT2/3 and TadA in the same orientation: **e** *Tb*ADAT2/3 with tRNA cytosine-base shown; **f** apo-*Mm*ADAT2/3 (PDB: 7NZ7), with the conserved H298$^{MmADAT3}$ as the only ADAT3 residue present in the active site; **g** bacterial *Sa*TadA (PDB: 2B3J) with nebularine bound, with M92$^{SaTadA*}$ residue contributed by the second protomer of the homodimer. Residue side chains colored by heteroatom with zinc atoms in yellow.

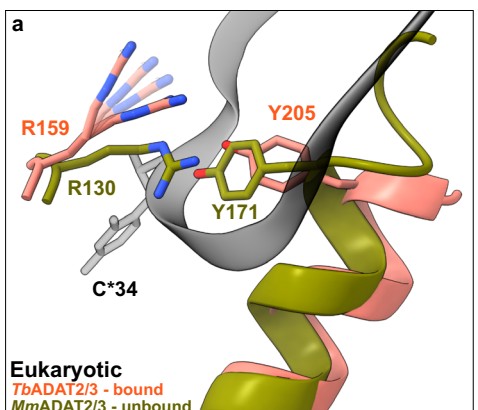

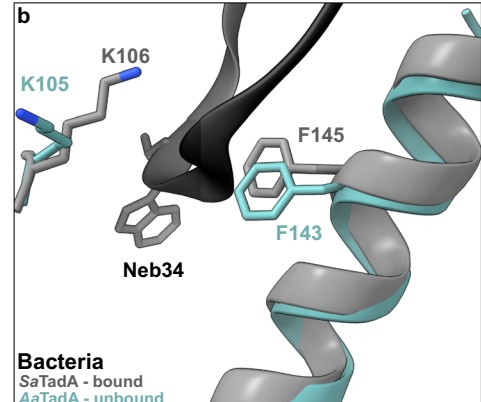

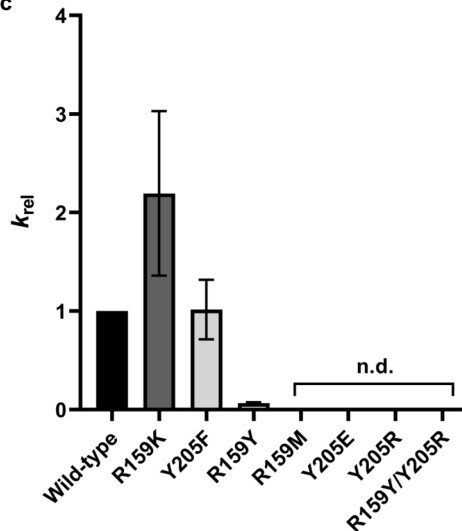

**Fig. 3 | The tRNA anticodon loop passes through a molecular RY-gate in ADAT2.**
**a** Superposition of the tRNA-bound gate in $Tb$ADAT2/3 and the apo-gate in $Mm$A-DAT2/3 (PDB: 7NZ7). The side chain of R159$^{Tb\text{ADAT2}}$ is represented as flexible due to its poor side chain density. **b** The corresponding positions in bacteria deaminase TadA, RNA bound versus unbound (PDB: 2B3J and 1WWR, respectively). **c** Fold change of the observed rate constants of mutants to wild-type ($k_{rel} = k_{obs}$ mutant/$k_{obs}$ WT). Graph values calculated from three independent replicates. All mutants are of $Tb$ADAT2 co-expressed with $Tb$ADAT3. n.d. denotes no detectable activity after 24 h assay incubation. $k_{obs}$ values obtained from single-turnover kinetic assays (n≥3). Source data are provided as a Source Data file.

protein surface[24]. Since base-specific recognition does not take place in ADAT2/3[37,41], we looked for other features that could evoke such a non-canonical anticodon loop rearrangement. We identified a eukaryote-specific molecular "RY-gate" formed by residues arginine R159$^{Tb\text{ADAT2}}$ and tyrosine Y205$^{Tb\text{ADAT2}}$, which may provide a certain control mechanism for the tRNA to overcome before fully entering the catalytic pocket (Fig. 3). In the RNA-free $Mm$ADAT2/3 structures (PDB: 7NZ7, 7NZ8), this gate is closed through a cation-π interaction between these residues, R130$^{Mm\text{ADAT2}}$ and Y171$^{Mm\text{ADAT2}}$ (Fig. 3a and Supplementary Fig. 3f). In the tRNA-bound $Tb$ADAT2/3 structure, the gate arginine R159$^{Tb\text{ADAT2}}$ is released, and appears to be flexible as judged by the map density (Supplementary Fig. 3d). Opening of this gate also frees the side chain of the second gate residue Y205$^{Tb\text{ADAT2}}$, which together with nucleobase C$_{32}$, can sandwich and internalize nucleotide U$_{36}$ via π-π stacking. This is likely to be a major driver in remodeling the anticodon loop (Supplementary Fig. 3e). Interestingly, despite the presence of an equivalent positively charged-aromatic pair (e.g., K106$^{Sa\text{TadA}}$ and F145$^{Sa\text{TadA}}$), this bacterial "gate" does not seem to operate because of suboptimal interacting geometry, which maintains these residues in an open conformation regardless of the presence or absence of RNA substrate; all available bacterial TadA structures in the PDB to date show an open configuration of these residues (Fig. 3b)[21,24,47,48]. In an earlier study, the substitution of R159$^{Tb\text{ADAT2}}$ to alanine displayed unaltered RNA-binding properties[39], so next we set out to scrutinize

the importance of the gate residues in catalysis through deamination assays. All amino acid substitutions that removed the aromatic nature of Y205$^{Tb\text{ADAT2}}$ (namely Y205E and Y205R) had no detectable activity after 24 h incubation, while a conservative mutation that maintains the aromaticity (Y205F) showed a reaction rate comparable to the wild-type enzyme. The necessity of the aromatic residue in this position indicates its importance in the base-sandwich for anticodon loop distortion. Next, we substituted the positive charge of the cationic gate residue R159$^{Tb\text{ADAT2}}$ with a hydrophobic methionine (R159M), which led to an inactive enzyme. However, replacing R159$^{Tb\text{ADAT2}}$ with tyrosine did not inactivate the enzyme but led to a 10-fold reduction of activity, indicating the importance of a positive charge for this gate residue. An "inverted gate" double mutant (R159Y, Y205R) was also inactive. A mutant mimicking the bacterial situation with a R159K substitution, conserving the positive charge but altering its geometry, strikingly, showed a more than 2-fold improved reaction rate compared to the wild-type, suggesting a lysine in this position is more efficient for deamination activity (Fig. 3c, d, Supplementary Fig. 3j, k, Supplementary Table 2). Once it has entered the gate, the anticodon loop gains access to the deep joint anticodon binding pocket in order to establish further interactions with the ADAT2 core: asparagine N157$^{Tb\text{ADAT2}}$ can form a hydrogen bond with the hydroxyl of the U$_{33}$ ribose and the splayed-out nucleotide U$_{33}$ can interact with ADAT2 residues 155$^{Tb\text{ADAT2}}$–158$^{Tb\text{ADAT2}}$ as well as glutamine Q202$^{Tb\text{ADAT2}}$ (Fig. 2d).

An ADAT3 RNA-binding loop, 328–333$^{TbADAT3}$, conserved between TadA and ADAT3 but not ADAT2, establishes contacts with the tRNA backbone at nucleotides 38–40 and the nucleobase A$_{38}$ via two histidines (H332$^{TbADAT3}$ and H333$^{TbADAT3}$) and an asparagine N331$^{TbADAT3}$, respectively (framed teal in Fig. 1b, Fig. 2d, Supplementary Fig. 3g–i). In conclusion, we identify a molecular gate formed by a cation-π interaction in eukaryotic ADAT2 which needs to be penetrated by the tRNA; the opened gate allows stabilizing aromatic stacking interactions with the C$_{32}$ and U$_{36}$ nucleobases to facilitate access to the deep anticodon loop binding cleft where further favorable protein–RNA interactions are established, altogether promoting the unusual anticodon loop configuration.

## The ADAT2 C-terminus embraces the anticodon-stem loop

The available crystal structures of apo-ADAT2/3 failed to resolve the very C-terminal portion of ADAT2 (ADAT2$^C$, residues 207–225$^{TbADAT2}$), a segment which is absent in bacterial TadA proteins. This region carries a conserved, positively charged motif at the C-terminal end of ADAT2$^C$, herein referred to as "KR-motif" (residues 216–220$^{TbADAT2}$ KRKRK) (Fig. 1b). The KR-motif was previously identified as essential for tRNA binding and deamination[39]. We fitted ADAT2$^C$ into the cryo-EM map guided by the AlphaFold model of the complex; following the C-terminal helix which is the last element resolved in the crystal structures, a conserved proline induces a turn of the polypeptide, such that the KR-motif bends back towards the helix (Fig. 4, Supplementary Fig. 1g). ADAT2$^C$ displays extensive contacts with the anticodon loop and stem: residues 207–209$^{TbADAT2}$ and 210–213$^{TbADAT2}$ of TbADAT2$^C$ embrace the flipped-out nucleobases G$_{35}$ and A$_{37}$ (Fig. 2d). Furthermore, the positively charged C-terminal KR-motif, which aligns into the major groove of the anticodon loop, likely provides extended interactions to the phosphate backbone of the anticodon stem towards nucleotides 28–31 (Fig. 4a). The less clear density here might reflect a certain fluidity of the interaction and adaptation to different sequence registers could be imagined, which would be required to adjust to the different types of tRNA. Taken together, ADAT2$^C$ including the essential KR-motif seems to be intrinsically flexible in the absence of tRNA, but can reorganize and adjust to bind into the major groove of the anticodon loop and also establish interactions with portions of the anticodon stem.

## ADAT3$^N$ binds the tRNA along the anticodon arm and elbow region

Experimental evidence has suggested a role of the eukaryotic ADAT3 N-terminal domain in RNA binding, which we seek out to elucidate in

detail through the cryo-EM structure. AlphaFold and RoseTTA models[44,49] of ADAT3$^N$ reveal a domain containing a slightly twisted 4-stranded antiparallel β-sheet with two α-helices in a βαββαβ arrangement (residues 13–140$^{TbADAT3}$) and an additional appended α-helix (residues 141–164$^{TbADAT3}$). This constellation closely resembles the crystal structures of the mouse and yeast homologues, with the main difference being a kinetoplast-specific insertion (residues 57–99$^{TbADAT3}$) that is predicted to be disordered (Fig. 5a, Supplementary Figs. 1e and 4a). Snapshots collected from previous crystal structures placed this domain in various positions with respect to the deaminase domains, suggesting that this domain is flexibly attached to the catalytic core. However, all previously experimentally observed positions of ADAT3$^N$ disagree with our cryo-EM map or would clash with other parts of the model (Supplementary Fig. 6h). The ADAT3 N-terminal domain is connected to the protein core between two linkers (N-terminal: residues 4–12$^{TbADAT3}$; C-terminal: residues 165–172$^{TbADAT3}$) which form together a quasi-parallel double linker and are well-resolved in our maps (Fig. 1d). Since the cryoSPARC map was not sufficient to fit the AlphaFold model, we generated a DeepEMhancer map and jiggle-fitted the ADAT3$^N$ without further side chain refinement (Supplementary Fig. 4b). In the model we identify two main positively charged contact regions between the ADAT3$^N$ and the tRNA (Fig. 5a). The first interaction region (spot 'A' in Fig. 5a) represents an elongated positively charged stretch facing the minor groove of the D-stem of the tRNA, likely binding the phosphate backbone via charged interactions. This region comprises the appendix α-helix of ADAT3$^N$ together with the C-terminal linker that connects to the deaminase domain. A second interaction hotspot can be observed in the interface between the ADAT3$^N$ domain and portions of the D-loop and elbow of the tRNA, in particular in the region of nucleotides 19 and 56, a base pair that is fundamental to the 3D structure of any tRNA (spot 'B' in Fig. 5a)[50,51]. Since the maps were not of sufficient quality to unambiguously identify the contact residues, we assessed single point mutants of ADAT3$^N$ of positively charged residues in either of the observed patches for their RNA-binding properties. Indeed the ADAT3 mutants K48A, R52E, K164E and R166E weaken the interaction of the ADAT2/3 complex with tRNA in electromobility shift assays (EMSA) (Supplementary Fig. 4c, d). Additionally, previously reported double and triple mutations of ADAT3 in this region have been attributed a role in tRNA binding and deaminase activity. (Fig. 5b, Supplementary Fig. 4a and Supplementary Table 3)[40,41]. The earlier observation that long variable arms in tRNA$^{Val}$ or tRNA$^{Ser}$ do not affect inosine formation is in accordance with our data as the map does not show density in the region where the variable

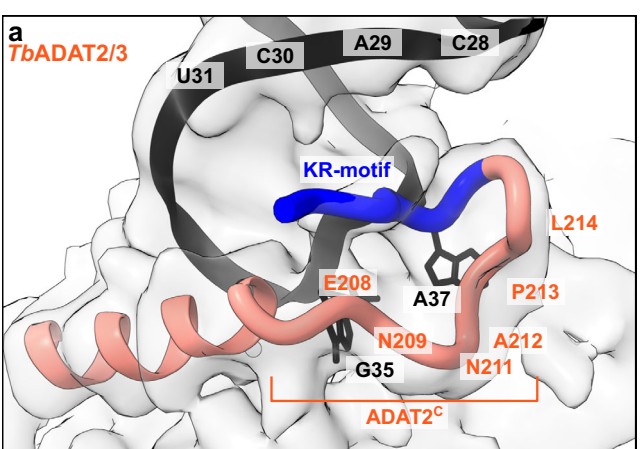
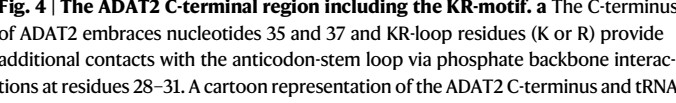
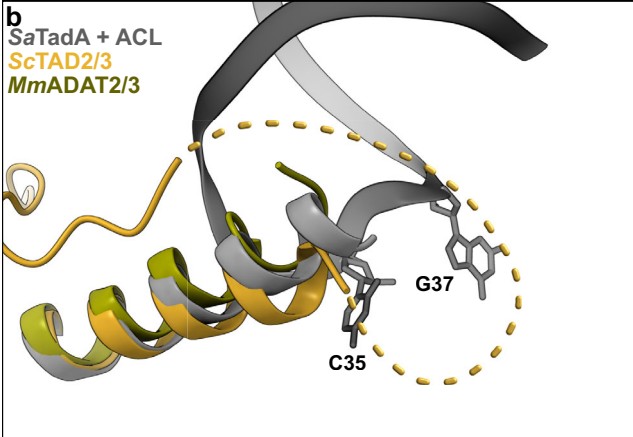

**Fig. 4 | The ADAT2 C-terminal region including the KR-motif. a** The C-terminus of ADAT2 embraces nucleotides 35 and 37 and KR-loop residues (K or R) provide additional contacts with the anticodon-stem loop via phosphate backbone interactions at residues 28–31. A cartoon representation of the ADAT2 C-terminus and tRNA

is shown with the final non-b-factor-sharpened cryoSPARC map. Due to the limited resolution, we refrain from showing side chains in the ADAT2$^C$ model. **b** The same region of ScTAD2/3 (PDB: 7BV5), MmADAT2/3 (PDB: 7NZ7), and RNA bound SaTadA (PDB: 2B3J). The flexible unmodeled loop of ScADAT2 is shown as a dotted line.

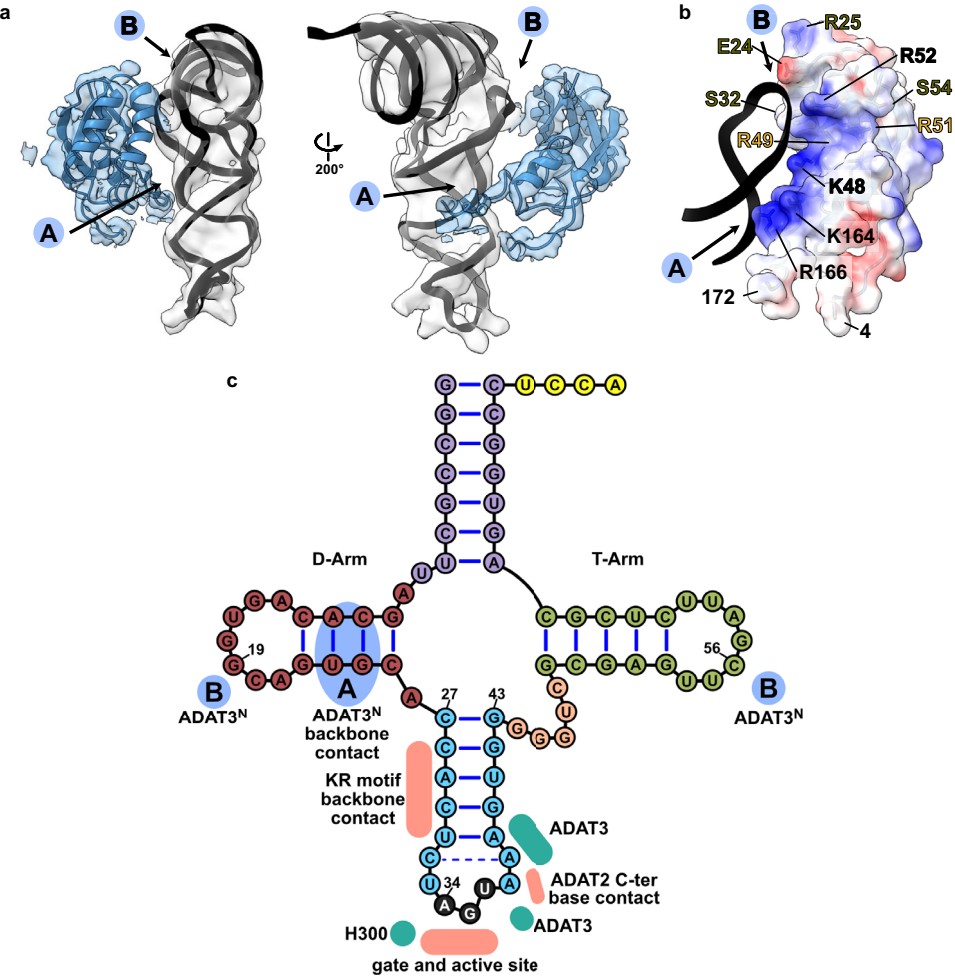

**Fig. 5 | ADAT3 N-terminal domain probes the three-dimensional structure of the tRNA. a** Two views of the ADAT3[N] fitted in the post-processed EM map, in blue, with 'A' and 'B' regions as the two contact points of our model with tRNA, as indicated in C. **b** ADAT3[N] including linkers (residues 4–172); surface model colored by vacuum electrostatic potential, positive charges in red and negative charges in blue, with D-Arm of tRNA in a black cartoon representation. Residues identified to play a role in tRNA binding are annotated in black, in olive from[41] and in yellow from[40]. (**c**) Schematic of the tRNA with interaction contacts. Anticodon-stem loop interactions are represented as a summary of the ones indicated in Fig. 2D. The two ADAT3[N] contact points, 'A' and 'B' are indicated in blue as above.

arm would be positioned[37,39,52,53]. We reasoned that the elongated, continuous positive patch is putatively able to bind RNA via phosphate backbone interactions independently of sequence context. The somewhat lower resolution of ADAT3[N] with respect to the deaminase core may indicate several possible binding modes and potential for adaptation to different tRNA molecules. The fact that ADAT2/3 is inactive on truncated anticodon loop substrates points towards the requirement of these extended interactions with ADAT3[N] for correct insertion of the tRNA into the active site[22,37]. In conclusion, our structure reveals a critical contribution of the ADAT3 N-terminal domain in the recognition of intact tRNA architectures via electrostatic contacts along the anticodon arm and in the elbow region. The human disease mutation V128M maps to ADAT3[N] (V139 in *Tb*ADAT3) and seems not in direct contact with the tRNA. This observation agrees with an earlier study showing that the mutant retained its tRNA binding properties, but showed subtly diminished levels of deamination most probably due to a slightly altered overall geometry[41].

## Discussion

The cryo-EM structure of the *Trypanosoma brucei* tRNA-bound ADAT2/3 complex pinpoints how the various RNA-binding motifs of the eukaryotic deaminase interact with the full-length tRNA through a concerted mechanism. The structure, combined with available structural and biochemical data allows us to identify key conformational

rearrangements that eukaryotic wobble base tRNA deaminases must undergo to correctly place the substrate in their active site for productive deamination. This allows us to propose the following improved model for a multi-step mechanism of tRNA recognition by ADAT2/3 (Fig. 6). In the free enzyme form, the N-terminal RNA-binding domain of ADAT3 (ADAT3[N]) is flexibly linked to the deaminase core, and the C-terminus of ADAT2 (ADAT2[C]) containing the positively charged RNA-binding KR-motif is unfolded (Fig. 6a)[40,41]. In accordance with prior propositions, we suggest an initial binding step in which RNA is promiscuously captured via electrostatic interactions by these two RNA-binding domains. Mutation or deletion of either the charged KR-motif of ADAT2[C 39] or putative RNA-binding residues in ADAT3[N 41], lead to tRNA binding defects that point towards an initial and simultaneous RNA binding of both regions (Fig. 6b). Subsequently, we imagine that the anticodon loop is progressively directed into the deaminase active site, guided by the expansion of favorable protein–RNA contacts between the positively charged stretch along the linker region with the anticodon arm, leading to insertion of the anticodon loop into the catalytic pocket. Throughout the insertion, ADAT3[N] and ADAT2[C] might survey the integrity of the accurate tRNA structure determinants, with non-tRNA structures being rejected due to the lack of the correct contacts. Once validated as any tRNA carrying the correct geometry, the substrate can then enter through the molecular RY-gate, formed by cation-π interactions between the gate

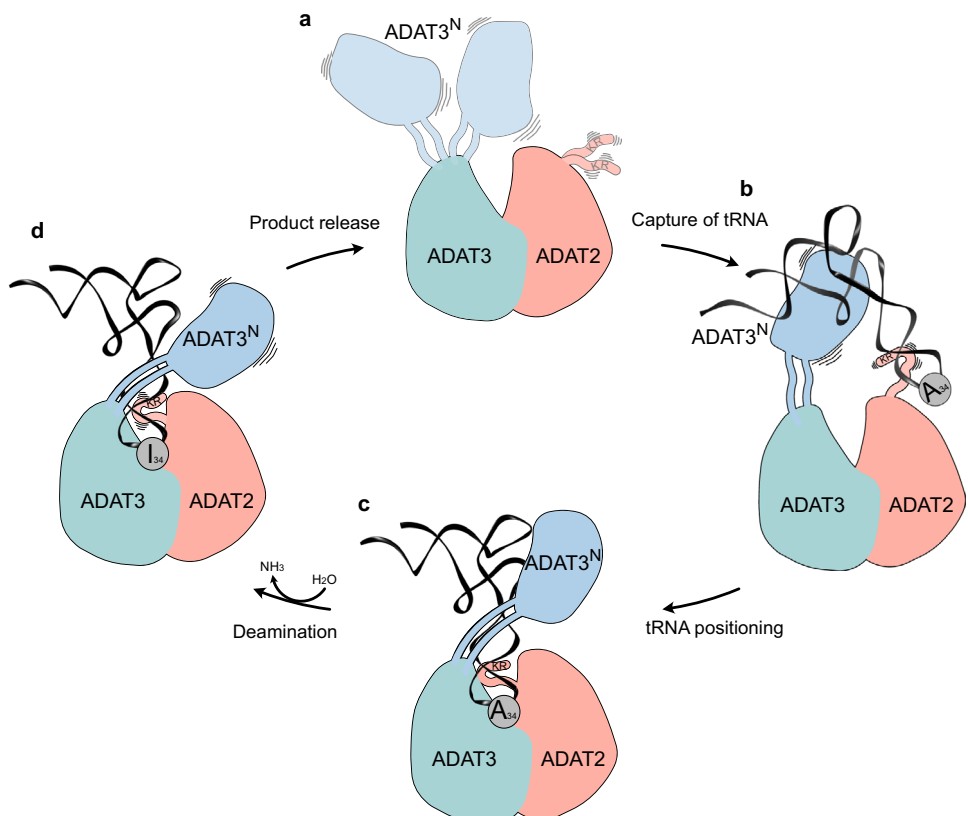

**Fig. 6 | The hypothetical multi-step tRNA recognition mechanism of ADAT2/3.** **a** Ligand-free enzyme state: in the free ADAT2/3 heterodimer ADAT2$^C$ is disordered and ADAT3$^N$ is flexibly linked. **b** Initial capture: the RNA recognition motifs in ADAT3$^N$ and in ADAT2$^C$ capture RNA; tRNAs are progressively guided by the extension of favorable interactions towards the anticodon binding cleft; hereby non-tRNA architectures will be rejected. **c** Anticodon loop insertion into the active site: The anticodon loop enters the active-site cleft through the RY-gate, and ADAT2$^C$ folds into the major groove of the anticodon loop to position the ligand for the reaction. **d** Substrate release: Detachment of the individual RNA-binding motifs initiates tRNA release.

residue side chains, into the anticodon binding cleft of the joint ADAT2/3 deaminase core. Supposedly, other cellular, non-tRNA 7-nucleotide RNA hairpins, resembling an anticodon loop, would not be able to overcome the gate in lack of the necessary contacts to ADAT3N. Base stacking with the released gate residue tyrosine Y205$^{TbADAT3}$ is enabled, allowing the nucleotides 33–36 to establish multiple interactions within the deep catalytic pocket. With the inserted anticodon loop, the unstructured portions in the ADAT2 C-terminus, notably the positively charged and essential KR-motif, adopt a folded conformation and help to correctly place the tRNA in a manner that is amenable for catalysis (Fig. 6c). At this point, non-cognate tRNAs (i.e. all tRNAs with G, U, or C in position 34) are eventually released, but for cognate tRNAs, the L-shape architecture, anticodon loop geometry, and the A$_{34}$ nucleotide will perfectly align to the three main binding motifs in ADAT2/3, namely ADAT3$^N$, ADAT2$^C$ and joint deaminase core, to trigger the deamination reaction. We suggest that for substrate release, Brownian motions of each individual RNA-binding site might initiate the discharge of non-cognate tRNAs (Fig. 6d). With hindsight, our structure may represent a transition state preceding product release with ADAT3$^N$ and ADAT2$^C$ showing hints of detached conformations, reflected in the reduced map resolution of both domains. We reason that when A$_{34}$ is present, tRNA turnover is accelerated by the successful catalysis, leading to a fast product release. This is supported by the >1 μM dissociation constant observed with an A$_{34}$-containing tRNA in previous binding studies as compared to ~180 nM observed with the G$_{34}$-containing tRNA, a non-substrate which is inert to the reaction[39]. Conversely, non-A$_{34}$-containing tRNAs, in the absence of catalysis, exhibit longer dwell times, leading to the conclusion that product release is driven by the energetics of the deamination reaction. Hence,

a non-A$_{34}$-containing tRNA may act as a competitive inhibitor for the reaction, which raises the question, how the cytoplasmic enzyme can be active in vivo in a pool of potential non-substrate inhibitors. Here, we can only speculate that the effective concentration of free A$_{34}$-tRNAs could possibly make the difference, as non-substrate tRNAs may be sequestered from the available cytosolic pool by other tRNA targeting enzymes, like tRNA synthetases, translation factors and translating ribosomes. To conclude, we propose mechanistically, that for productive A$_{34}$ deamination by eukaryotic ADAT2/3, all key enzyme-tRNA contacts, i.e., ADAT3$^N$, ADAT2$^C$ and the heterodimeric deaminase core, act in a concerted manner to capture the tRNA, probe for the correct geometry, and finally correctly position the substrate in the active site to trigger the deamination reaction.

Further, we reflected how the asymmetry of the ADAT2/3 heterodimer led to diversification of its tRNA targets. Several structures of the bacterial deaminase TadA, combined with biochemical analysis suggested that the bacterial enzyme makes sequence-specific contacts in their active site to efficiently target its single-substrate tRNA$^{Arg}_{ACG}$. It has been proposed that substrate diversification to multiple tRNAs by ADAT2/3 was the result of a gene duplication of the ancestral ADAT2 protein yielding ADAT3. The heterodimer could now acquire active-site mutations that effectively relaxed the need for sequence-specific recognition and would allow for the acquisition of tRNA binding domains distal from the active site (i.e., ADAT2$^C$ and ADAT3$^N$)[8,17,19,22,37,54]. However, in the tRNA-bound structure we do not observe evidence for a relaxation of the active site, but a rather high conservation to the ancestral enzyme. A rather rigid central pocket for the A$_{34}$ is observed when comparing it to the free enzyme structures. Furthermore, the extended anticodon binding pocket of both, the bacterial

TadA homodimer and the ADAT2/3 heterodimer, display similar ways of interaction with the splayed-out anticodon loop bases $C_{32}$-$A_{38}$, with no explicit differences explaining their different substrate preferences. However, as a major difference, the eukaryotic enzyme has acquired additional domains contributing multiple additional sequence-nonspecific interactions along the anticodon arm up to the elbow region of the tRNA (Fig. 5c and Supplementary Fig. 5). The appearance of these new interaction motifs potentially lead to a multi-step binding mechanism and may have reduced the dominance of the active site for tRNA binding. Thus, jointly all RNA-binding elements in the heterodimer may even provide sufficient affinity to guide substrates with imperfect sequences into the active site, as long as they satisfy the need for an overall tRNA geometry. As a relic of its origin, in extant ADAT2/3 enzymes, the ADAT2 enzyme is still able to form homodimers in vitro, yet these are inactive[41,55]; PDB: 3DH1), reflecting the evolved dependency on ADAT3 for enzyme activity.

The tRNA-bound cryo-EM structure of the ADAT2/3 heterodimer presented here allows us to unambiguously pinpoint the exact function of all crucial tRNA binding features that differentiate ADAT2 and ADAT3. ADAT2 specific features are the RY-gate, necessary for access to the deaminase active site and the essential C-terminal KR-motif, which, intrinsically unfolded in the ligand-free state, contributes to the correct placement of the anticodon-stem loop once the tRNA is bound. ADAT3, on the other hand, has acquired an indispensable additional N-terminal RNA-binding domain to probe global tRNA geometry through extended sequence-independent contacts to the tRNA phosphodiester backbone; a somewhat similar case has been observed for the tRNA (guanine(37)-N1)-methyltransferase Trm5, that also recognizes the full tRNA L-shape via its "sensor" D1 domain, which probes for the intact 19–56 base pair before recruiting the "effector" catalytic domain[56–58]. In a similar manner, also the tRNA anticodon modifying enzymes TilS and TiaS recognize features of their cognate tRNA via domains which are distal from the active center, albeit base specific in these cases[59]. Interestingly, while the evolutionarily conserved $A_{34}$ pocket in ADAT2/3 appears of relatively rigid character, the distal eukaryote-acquired RNA-binding features show a certain flexibility either via connection through linkers (ADAT3[N]) or induced-fit adaptation of the protein portion (ADAT2[C]). This flexibility is most likely the key to the accommodation of multiple tRNAs with divergent anticodon loop sequences (Supplementary Fig. 5). Furthermore, we find it interesting that ADAT3s of different eukaryotic clades all have evolved different loops to obstruct their pseudo-active site, potentially to prevent tRNAs to bind in an unproductive manner (Supplementary Fig. 6)[40,41].

In conclusion, the cryo-EM structure of the full-length ADAT2/3 heterodimer bound to a tRNA delivers the long-sought rationale of how a sequence-specific single-substrate wobble base deaminase has evolved into a multi-tRNA acceptor. Key to its evolution were the acquisition of additional RNA-binding domains and motifs (ADAT3[N] and ADAT2[C]), and the development of a multi-step tRNA recognition mechanism, which was only possible after gene duplication allowing the two subunits to diverge and undergo neo-functionalization. The acquired distal RNA-binding motifs provide additional ductile affinity to the ligand through induced-fit mechanisms, but moreover they serve to authenticate tRNAs via their global 3D features; delicately counterbalancing multi-sequence recognition while keeping the enzyme's mutagenic potential in check.

## Methods
### Protein expression and purification
For structure determination full-length ADAT2 (Tb927.8.4180, with two point mutation that improve solubility and reduced ADAT2 homodimerization: A87G/C117S) and full-length His-tagged ADAT3 (Tb927.11.15280), and mutants of ADAT3 used in tRNA binding (K48A, R52E, K164E, and R166A), were co-expressed in Hi5 cells using the

multibac and Bigbac insect cell expression systems[60,61]. Cells were resuspended in lysis buffer (20 mM Tris pH 7.5; 200 mM NaCl; 20 mM Imidazole; 2% Glycerol; supplemented with 1 mM of PMSF, 2 μg/mL of DNase and 25 μg/mL of RNase) and lysed by sonication (5 min, 30% Amplitude, 5 s on, 5 s off; Vibra-cells, Sonics). The lysate was clarified by centrifugation (40,000 × g, 30 min) and loaded into a 5 ml HisTrap HP affinity chromatography column (Cytiva), that was washed with 40 column volumes of high salt buffer (20 mM Tris pH 7.5; 1 M KCl; 50 mM Imidazole) and 40 column volumes of low salt buffer (20 mM Tris pH 7.5; 200 mM NaCl; 50 mM Imidazole). The sample was eluted with elution buffer (20 mM Tris pH 7.5; 200 mM NaCl; 600 mM Imidazole) and flushed through a HiTrap Q HP 5 mL anion exchange chromatography column (Cytiva) to remove charged impurities. After 3 C protease tag cleavage, the sample was concentrated through centrifugation in Centricon filters and finally purified through size-exclusion chromatography on a HiLoad 16/600 Superdex 200 column (Cytiva) pre-equilibrated with SEC buffer (20 mM Hepes pH 7.5; 100 mM NaCl; 2 mM DTT). All steps were performed at 4 °C.

For biochemical assays, TbADAT2/3 was recombinantly expressed in E. coli as previously described. Briefly, the coding sequences of TbADAT2 (A87G/C117S) and TbADAT3 were cloned into expression vectors pETDuet-1 and pET-28(a)+ and used to transform E. coli BL21-IRL strain. For expression, a 10 mL overnight culture was added to 1.5 L of prewarmed 2XYT media and grown at 37 °C to an OD600 of 0.6–0.8. Recombinant protein expression was then induced with isopropyl β-d-1-thiogalactopyranoside (IPTG, 0.5 mM final concentration) overnight at 25 °C. The following procedures for protein purification were carried out at 4 °C. Cells were pelleted, suspended in lysis buffer (20 mM Tris pH 8.0, 1 M NaCl, 5 mM β-mercaptoethanol, 20 mM Imidazole, protease inhibitor cocktail and 0.1% NP40) and lysed by sonication with a Branson Sonifier 450 (three times, 30 s intervals, 30 s rest between sonication). The resulting lysate was clarified by centrifugation at 40,000 × g for 30 min followed by a second centrifugation step at 55,000 × g for 30 min. Clear lysate was collected and loaded onto a 1 mL HisTrap HP column (Cytiva, $Ni^{2+}$-nitrilotriacetic acid column). The column was washed with the lysis buffer followed by a lower salt wash containing 20 mM Tris pH 8.0, 100 mM NaCl, 5 mM β-mercaptoethanol, and 50 mM Imidazole. The bound protein was eluted with 600 mM Imidazole. Peak fractions were pooled then passed through a PD-10 desalting column (Cytiva) and stored as aliquots at −80 °C in buffer containing 100 mM Tris pH 8.0, 100 mM NaCl, 0.5 mM $MgCl_2$, 0.2 mM EDTA, 2 mM 1,4-dithiothreitol, and 15–20% glycerol.

### Thermostability assay
Thermal stability of TbADAT2/3 was determined by differential scanning fluorimetry (DSF). Briefly, TbADAT2/3 (in a final concentration of 4 μM) was mixed with SYPRO Orange in DSF buffer (20 mM Hepes pH 8; 200 mM NaCl; 2 mM DTT) and incubated in a temperature gradient for denaturation. The melting temperature was approximated to the midpoint between maximum and baseline signals of the SYPRO Orange fluorescence[62].

### ADAT2/3 ligand sequencing
Nucleic acids co-eluted in the size-exclusion peak-fraction of recombinantly expressed TbADAT2/3 derived from E. coli as indicated by an absorption 260/280 ratio of ~1.7. Nucleic acids were phenol-chloroform extracted and precipitated with sodium acetate. The sequencing reaction was prepared following a tRNA sequencing protocol with the NEXTflexTM Small RNA-Seq Kit v3[63]. Adapter removal, 5' and 3' trimming of the result sequences and removal of sequences <15 bases resulted in 3577188 reads which were aligned with bowtie2 to an E. coli or E. coli tRNA index, respectively, and resulted in a 57.69% or 25.96%, respectively, overall alignment rate.

## tRNA synthesis for cryo-EM

The *Trypanosoma brucei* threonine tRNA (*Tb*tRNA$^{Thr}_{CGU}$) sequence was cloned in a pUC19 vector, between a T7 promoter and a BstNI cleavage site. The vector was linearized with BstNI and used as a template for in vitro transcription. The reaction was carried out in T7 reaction buffer (40 mM Tris pH 8; 5 mM DTT; 1 mM spermidine; 0.01% Triton X-100; 30 mM MgCl$_2$), with 4 mM of each NTP. For a 10 mL reaction, 0.5 mg of linear DNA and 0.4 mg of T7 RNA polymerase were used. The reaction was incubated at 37 °C for 8 h. The transcribed RNA was iso-propanol precipitated, Urea-PAGE purified and desalted following standard procedures. The tRNA$^{Thr}$ was refolded by heating at 95 °C for 10 min in 10 mM Tris pH 7.5 and 50 mM NaCl, slow cool down and adding MgCl$_2$ to 1 mM when it reached 37 °C. After refolding, *Tb*tRNA$^{Thr}_{CGT}$ was further purified by size-exclusion chromatography, in a Superdex 200 Increase 10/300 GL size-exclusion chromatography column (Cytiva) pre-equilibrated with refolding buffer (10 mM Tris pH 7.5 and 50 mM NaCl, 2 mM MgCl$^2$).

## Cryo-EM sample preparation

For complex formation, ADAT2/3 heterodimer and tRNA were mixed and diluted in interaction buffer (20 mM Hepes pH 7.5; 50 mM NaCl; 2 mM MgCl2; 0.5 mM TCEP) to a final concentration of 32 μM and 35 μM, respectively. Excess tRNA was removed through size-exclusion chromatography on a Superdex 200 Increase 3.2/300 column (Cytiva). UltrAuFoil R 1.2/1.3 300 Au mesh (Quantifoil) grids were glow discharged with residual air for 30 seconds, on each side in a PELCO easiGlow device operated at 30 mA. Two microliters of sample were applied in each side of the grid before blotting for 3 s (blot force 0) and vitrified in liquid ethane using a Vitrobot MARK IV (Thermo Fisher Scientific) operated at 4 °C and 100% humidity.

## Cryo-EM Data acquisition

Micrographs were collected with a 300 kV Titan Krios (Thermo Fisher Scientific) electron microscope, equipped with a K2 Summit direct electron detector and a GIF Quantum energy filter (Gatan). Data were acquired using serialEM[64] at a nominal magnification of 165,000, resulting in a pixel size of 0.81 Å. For a better sampling of particle orientations, the stage was tilted 30°; to reduce beam induced motion, the beam size was increased to 650 nm. Movies were acquired for 10 s at an electron exposure rate of 3 e/pixel/s, resulting in a total electron exposure of 55.00 e/A$^2$ at the sample level, fractionated into 80 movie frames. Three movies were acquired per hole, for a total of 6145 movies, with a defocus range from −1.0 to −1.8 μm.

## Cryo-EM data processing

Movies were motion corrected (using a 5 × 5 patch model) and patch CTF estimated (3 × 3 patch model) in Warp. A total of 1,156,506 particles were picked and extracted (300 pixel box) from the corrected micrographs using the BoxNet2Mask_21080918 model[42]. After initial 2D classification in CryoSPARC[43], we could identify three main particle subsets: tRNA, with 62,505 particles; ADAT2/3 heterodimer, with 83,218 particles; and ADAT2/3-tRNA complex, with 559,713 particles. One round of 3D classification generated three classes using ab initio reconstruction, followed by hetero refinement. The 427,165 particles from the main class were 2D classified and blurred 2D classes were discarded. The remaining 379,795 particles were once more 3D classified by generating 4 ab initio reconstructions followed by hetero refinement. The 105,718 particles from the best class were locally CTF refined, and further refined in a non-uniform refinement to generate the final 3.6 A map that was post-processed using DeepEMhancer[45].

## Model generation with AlphaFold

The model used to build the ADAT2/3 structure in the EM map was generated with AlphaFold. The sequences of ADAT3 and ADAT2, connected via a 45-residue AGS linker, were submitted to the AlphaFold colab notebook, and the 5 resulting models were compared based on their predicted IDDT and PAE.

(https://colab.research.google.com/github/sokrypton/ColabFold/blob/main/AlphaFold2.ipynb#scrollTo=G4yBrceuFbf3).

## Model building

The final non-b-factor-sharpened cryoSPARC map was sharpened or blurred in Coot 0.9.4[65] for easier interpretation. For the model building, the deaminase domain of ADAT2 and ADAT3 were built based on a model of *Tb*ADAT3 calculated by Phyre$^2$ [66] using the *Mm*ADAT3 (PDB: 7NZ7)[41] as a template, and an AlphaFold generated model of *Tb*ADAT2/3. The AlphaFold generated N-terminal domain of ADAT3 was jiggle-fitted in the DeepEMhancer post-processed map using the traced linkers as extra restraints. For the tRNA, we rigid-body fitted the crystal structure of yeast phenylalanine tRNA (PDB: 1EHZ)[67] into the density map, mutate nucleotides to correspond to the *Tb*tRNA$^{Thr}_{CGU}$ sequence, and refined only the portion relative to the anticodon-stem loop bases 28–42. The model was then refined against the final, non-b-factor-sharpened cryoSPARC map in an interactive manner with Ramachandran, secondary structure, base pair and base stacking restraints, using both Coot 0.9.4 and Phenix[68]. Alignments were made with pymol or clustal, and represented with ESPript 3[69,70].

## Deamination assay

To prepare substrate for deamination assays, full-length tRNA$^{Thr}_{AGU}$ was in vitro transcribed using a T7 promoter and internally labeled with [α-$^{32}$P]-ATP as previously described[17,71]. Single-turnover kinetics were performed in reaction buffer containing 40 mM Tris pH 8, 5 mM MgCl$_2$, and 10 mM DTT. Enzyme was added to the reaction in excess to labeled tRNA substrate (1 nM) and incubated at 27 °C as previously described[39]. Aliquots from the reaction were taken at various time points up to one hour (with exception of R159Y$^{TbADAT2}$, which was up to four hours). Mutants with no detectable activity were incubated for 24 h. Samples were quenched via phenol extraction followed by ethanol precipitation. The pelleted samples were then resuspended and treated with Nuclease P1 overnight (18 h) at 37 °C. Digested samples were then dried with high heat in a SpeedVac DNA 110 concentrator system (Savant). Dried sample pellets were resuspended in water and spotted onto a cellulose thin-layer chromatography (TLC) plate (Merck). Products were resolved for 2 h in one dimension in solvent C containing 0.1 M sodium phosphate (pH 6.8): ammonium sulfate: n-propanol (100:60:2 [v/w/v]). The TLC was dried and exposed to a PhosphorImager Screen overnight. Results were visualized using Typhoon FLA 9000 (GE) and analyzed using ImageQuant software. Fraction of inosine formed was calculated using pI/(pA + pI). The fraction A$_{34}$ to I conversion was plotted against time and fit using MATLAB software to a single exponential curve [f = a(1 − e$^{-kt}$)], where f represents product formed, a denotes product formed at the end point of the reaction, k signifies k$_{obs}$, and t is time.

## Electrophoretic mobility shift assay

Refolded and SEC purified *Tb*tRNA$^{Thr}_{CGU}$ in a final concentration of 20 nM was incubated on ice with a serial dilution of ADAT2/3 for 30 min in binding buffer (20 mM Hepes pH 8; 50 mM NaCl; 2 mM MgCl$_2$; 2 mM DTT; 5 mM KCl). Glycerol was added to the samples to a final concentration of 10%, and the products were separated on a pre-run 6% nondenaturing polyacrylamide gel, run in 1× TAE at 100 volts for 1h30 at 4 °C. The gel was stained with SybrGold (Invitrogen) and imaged in ChemiDoc MP Imaging System (Bio-Rad). Unbound tRNA samples were quantified using ImageLab (Bio-Rad), and the binding data were fit using Origin 7.0 software.

## Circular dichroism spectroscopy

Circular dichroism (CD) spectra of wild-type and RY molecular gate mutants ADAT2/3 were measured at room temperature in 10 mM Tris pH 7.5 buffer, using an AVIV 62DS Spectropolarimeter. CD data were

collected at a scan rate of 1 nm/s; spectra reported represent the average of three scans.

## Reporting summary

Further information on research design is available in the Nature Research Reporting Summary linked to this article.

## Data availability

EM maps are available at EMDB: EMD-15690 and the model is deposited at PDB: 8AW3. Source data are provided with this paper.

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

## Acknowledgements

We acknowledge Martin Pelosse for support in using the Eukaryotic Expression Facility at EMBL Grenoble, and Sarah Schneider for support in using the EM Facility at EMBL Grenoble. This work benefited from access to the cryo-EM platform of the Structural and Computational Biology Unit at EMBL Heidelberg. We acknowledge the HTX Team for the thermostability assay. We acknowledge the EMBL Genomic Core Facility for the tRNA sequencing. This work used the platforms of the Grenoble Instruct-ERIC center (ISBG; UAR 3518 CNRS-CEA-UGA-EMBL) within the Grenoble Partnership for Structural Biology (PSB), supported by FRISBI (ANR-10-INBS-0005-02) and GRAL, financed within the University Grenoble Alpes graduate school (Écoles Universitaires de Recherche) CBH-EUR-GS (ANR-17-EURE-0003). We thank Aline Le Roy and Christine Ebel, for assistance and access to the Protein Analysis On Line (PAOL) platform. We thank the Kowalinski lab members for discussion and comments throughout the course of the project. We also thank Andrew McCarthy and Sebastian Falk for comments on the manuscript. The present work was funded in part by the National Institutes of Health Grant GM084065-11 to J.D.A. The work was supported by a grant from the French Agence Nationale de la Recherche to E.K. (ANR-20-CE11-0016).

## Author contributions

E.K. designed the study. E.K., L.G.D., and L.T. conducted experimental work. F.W. collected the tilted cryo-EM data. L.G.D. processed the cryo-EM data and built the model. L.G.D. and E.K. interpreted the data and wrote the manuscript. M.A.R. performed experiments and interpreted the data. J.D.A. interpreted the data and wrote and edited the manuscript. A.Z. performed experiments and interpreted the data.

## Funding

## Competing interests

The authors declare no competing interests.
