## [Peer Review File · Nature Communications]

Structural basis for sequence-independent substrate selection by eukaryotic wobble base tRNA deaminase ADAT2/3REVIEWER COMMENTS

Reviewer #1 (Remarks to the Author):

In this manuscript, Dolce et al. report the cryo-EM reconstruction of *T. brucei* ADAT2/3, the adenosine deaminase complex responsible for deamination of adenosine to inosine at the tRNA wobble base position, bound to a full-length tRNA. The structure revealed that the tRNA anticodon loop adopts a distorted conformation similar to what has been observed in the bacterial homologue TadA. In addition, several other domains of ADAT2/3 were found to make sequence-independent contacts with the full-length tRNA, explaining how eukaryotic enzymes are both specific for tRNA substrates and promiscuous enough to act on multiple tRNA targets.

Overall, this work advances our understanding of RNA substrate recognition by RNA modification enzymes. I recommend that the manuscript be accepted pending revisions as detailed below.

Major comments:

(1) The nominal resolution of the EM map is 3.6 Å, but the densities shown in the figures do not reflect that. It would be helpful for quality control purposes if the authors could include a panel in the supplement showing a high-resolution region of the model so that the overall density quality can be confirmed.

(2) It is not always clear which maps are displayed in each figure (e.g. Figs. 1 & 4), and which maps were used for modeling. For example, in figure 1c, it is stated that the “final cryoSPARC map” was used for modeling. Does this correspond to the non-b-factor-sharpened map (“map”) or the b-factor-sharpened map (“map_sharp”) output by the Cryosparc non-uniform refinement? From the “Model Building” section of the methods (page 29), it is unclear whether the DeepEMhancer post-processed map was used only for fitting of the ADAT3 N-terminal domain as it seems from the main text. What is the difference between the DeepEMhancer map and the final map referenced in the methods a few sentences later? Please clarify.

(3) Because DeepEMhancer processing could in theory cause artifacts that remain undetected when the ground truth is not known (as mentioned in the discussion of Sanchez-Garcia et al. 2021), it would be useful to validate the model fit using another approach. For example, is the ADAT3N fit qualitatively the

same when fit into the Cryosparc unsharpened map or a locally filtered version of the map (or another variation of the map in which the ADAT3N region is more visible)?

(4) Regarding the “RY-gate” (page 12), it’s not clear that one can interpret the “gating” activity to be important. An alternative explanation, consistent with the activity assays presented here, would be that the positive charge at position 159 is important for some other reason, such as interacting with the tRNA substrate as or after it enters the active site. The authors should address this in the text.

(5) How much adjustment of the AlphaFold model was necessary to fit the KR motif into the EM density? From Fig. 4 and the validation report, the fit of these residues is on the poorer side - rigid body fitting would seem reasonable there. Additionally, from the view shown in Fig. 4a, it isn’t clear whether the density extending out from the tRNA (marked with the blue arrow) is really due to the KR loop, given the lack of continuous density for the side chains. Is there another threshold that can be displayed (in this figure or the supplement) to make this convincing?

(6) In the “Cryo-EM Data Processing” section of the methods (page 28), the authors state “One round of 3D classification generated three ab-initio reconstructions”. Could the authors please rephrase this to clarify whether they used the “ab initio reconstruction” job within Cryosparc or the “3D classification” job?

Minor comments:

(1) Regarding the discussion of inhibition by non-substrate tRNAs on p. 20: This is an interesting aspect of this biology. Is anything known about levels of ADAT2/3 or the various tRNAs in the cytoplasm?

(2) Why weren’t the insect cell expressed ADAR2/3 proteins used for the biochemical assays? Do the solubility-increasing point mutations affect activity, or is there another reason?

(3) In the “Cryo-EM Data acquisition” section of the methods (p. 28): the “dose” should be more precisely referred to as “electron exposure rate” and “total dose” should be more precisely referred to as “total electron exposure”.

Reviewer #2 (Remarks to the Author):

Here Kowalinski and colleagues report the cryoEM structure of the *T. brucei* tRNA-specific adenosine deaminase ADAR2/3 heterodimer bound to a model substrate tRNA. ADAT2/3 selectively deaminates A34 in the anticodon loops for several eukaryotic tRNAs and is related to the bacterial TadA enzyme that carries out a similar reaction at A34 of tRNA(Arg) as a homodimer. While structures of bacterial TadA, free and bound to RNA, have been known for many years and its mechanism for substrate recognition is well established, how the eukaryotic ADAT2/3 heterodimer reacts with a diverse group of tRNAs that vary in their anticodon loop sequence was not known (prior to the work reported here). This structure and accompanying biochemical data constitute an important advance in our understanding of substrate recognition by RNA modifying enzymes in general and specifically how eukaryotes modify the A34 position of several different tRNAs. This is a well written paper. The structure both explains data already reported in the literature and generates new hypotheses for the mechanism of ADAT2/3 substrate recognition and processing. I have only the following one critical comment.

A key distinction between the bacterial system and the eukaryotic system described here is the observation of contacts between the ADAT3 component of the heterodimer and the D arm and T arm of the tRNA. The authors state that others have identified ADAT3 residues important for RNA binding and present a table summarizing those results (Extended data table 2) and a figure showing the locations of these residues (Fig. 5b). However, it appears their structure suggests that some of these residues may be involved in direct contact with RNA while others may not. More discussion is needed about what their structure says about what specific residues are likely involved in direct RNA contacts in these locations. If their data are insufficient to identify specific amino acid-nucleotide contacts in these regions, this should also be stated.

Reviewer #3 (Remarks to the Author):

In this manuscript, Dolce et al. have determined the structure of ADAT2/3 in complex with its full-length tRNA substrate revealing insight into the mechanisms of tRNA A-to-I editing in eukaryotes. Thereby, they visualize how the N-terminal domain of ADAT3 helps to recognize the entire tRNA forming interactions in particular with the D arm. Moreover, they identify a novel gate of two amino acid residues in the active site of ADAT2 that is closed in the RNA-free structure (published previously) but opens to accommodate the anticodon arm in the active site. The importance of these amino acid residues is supported by biochemical studies.

In general, the experiments are technically sound. The manuscript is overall well written and compares the new structure to previously reported structures of apo ADAT2/ADAT3 and the bacterial homolog TadA bound to an anticodon stem-loop.

While this manuscript provides new insight into the molecular details how ADAT2/3 recognize tRNA, the manuscript in its current format is could be further improved to broaden the scope and impact. In particular, several questions around ADAT2/3 are not clearly addressed by the authors as outlined below.

1. The authors should describe more clearly the current understanding how ADAT2/3 discriminate cognate from non-cognate tRNA. This is in particular important for readers who are generally interested in RNA recognition and modification, but who are not experts on A-to-I editing. Based on my own reading (e.g. HR Frigole et al, RNA 2019), it seems that ADAT2/3 recognizes and modifies all eukaryotic tRNA which have an A at position 34. Thus, it might not be required for ADAT2/3 to recognize any other structure- or sequence-elements in tRNA except for ensuring that a tRNA is bound rather than any other RNA with an anticodon-like loop. This is eluded to in the manuscript but can be stated much more clearly to set the stage for the discussion of tRNA recognition.

2. Why did the authors choose a non-cognate tRNAThr for the structural studies rather than transcribing a cognate tRNA with an A34 to C mutation? It cannot be rule out that such a cognate tRNA could be position slightly different on ADAT2/3.

3. The structure provides limited insight into the active site and catalysis since the catalytic glutamate is disordered and a C34 is used to prevent catalysis. Is it possible to generate a structure using a cognate tRNA substrate with an A34, but mutating the catalytic glutamate in ADAT2?

Or in other words, can their structure provide insight into catalysis? Or has this been previously deciphered based on the structure of bacterial TadA with RNA bound?

4. Interestingly, the authors claim that a key difference between bacterial TadA and eukaryotic ADAT2/3 lies in the recognition of the anticodon loop which is sequence-specific for TadA, but sequence-independent for ADAT2/3. To corroborate this claim, the authors should include and discuss additional figures highlighting the changes in ADAT2/3 compared to TadA that allow the recognition of each anticodon base without conferring sequence specificity.

5. When investigating the importance of the RY-gate, the authors make several single-residue substitutions. Do these substitutions affect protein stability and folding? The authors should assess this by melting curves or CD spectroscopy to show that the ADAT2/3 variants are correctly folded.

6. Have the corresponding substitutions of the RY gate been made in bacterial TadA and what are their effects?

7. Figure 3 c and d are redundant as the same information (krel as a representation of kobs) is shown twice.

8. To determine kobs, the authors must measure time courses where more than one data point is outside the endlevel (Extended Data Fig 3). In other words, there should be more data points in the first 10 min.

9. E. Ramos-Morales et al., NAR 2021, reported the structure of the ADAT2/ADAT3 complex in absence of RNA and predicted the interaction with tRNA (graphical abstract and Fig. 5). In this manuscript, the authors should discuss how their cryo-EM structure differs or confirms this previously published model which looks very similar in general. Here, the authors state that “all previously observed positions (of the ADAT3 N terminal domain?) disagree with our cryo-EM map”. This statement should be supported by a comparative figure.

10. In the introduction, the authors mention mutations in ADAT3 that cause a rare disorder. What insight does the ADAT2/3-tRNA structure provide regarding the impact of these mutations on ADAT2/3 structure and function? The authors should map these mutations onto the structure and discuss possible effects.

11. This structure unambiguously shows that ADAT2/3 can tightly bind to non-cognate tRNA. In the discussion, the authors speculate why this may not lead to competitive inhibition in the cell, but their argumentation is not convincing. As much as non-cognate tRNAs may be “sequestered from the available cytosolic pool by other tRNA targeting enzymes”, the same will hold true for cognate tRNAs. Thus, this structure raises almost more questions than answers regarding the discrimination of cognate and non-cognate tRNAs by ADAT2/3. To address this issue, the authors may conduct competition experiments in the presence of both cognate and non-cognate tRNAs to at least quantify whether competition takes place.

Dear reviewers,

All authors would like to express their gratitude to you for your precious time and energy taken to help to improve our manuscript and enhance its comprehensibility. We also thank you for your positive feedback and appreciation of our study.

You have given us valuable food for thought and following your constructive suggestions, the authors addressed all comments. The authors are truly convinced that the reviewers' input improved the scientific accuracy of the manuscript as well as its comprehensibility for the reader.

Thank you a lot,
Eva Kowalinski (in the name of all authors)

In the following we address all reviewers' comments:

Reviewer #1 (Remarks to the Author):

In this manuscript, Dolce et al. report the cryo-EM reconstruction of *T. brucei* ADAT2/3, the adenosine deaminase complex responsible for deamination of adenosine to inosine at the tRNA wobble base position, bound to a full-length tRNA. The structure revealed that the tRNA anticodon loop adopts a distorted conformation similar to what has been observed in the bacterial homologue TadA. In addition, several other domains of ADAT2/3 were found to make sequence-independent contacts with the full-length tRNA, explaining how eukaryotic enzymes are both specific for tRNA substrates and promiscuous enough to act on multiple tRNA targets.

Overall, this work advances our understanding of RNA substrate recognition by RNA modification enzymes. I recommend that the manuscript be accepted pending revisions as detailed below.

Major comments:

(1) The nominal resolution of the EM map is 3.6 Å, but the densities shown in the figures do not reflect that. It would be helpful for quality control purposes if the authors could include a panel in the supplement showing a high-resolution region of the model so that the overall density quality can be confirmed.

To address the concern, we recalculated the resolution estimation via different programs and added these values to the manuscript (page 7). Supplementary figures 3a, b, d, e show relevant densities.

Initial: "Data was processed with a combination of available software suites resulting in a final map with a resolution up to 3.6 Å, that is reasonable with respect to particle size"

Rephrased to: "Data was processed with a combination of available software suites, and the resolution of the final map was calculated by the gold standard fourier shell correlation method using the cutoff of 0.143 by four programs: cryoSPARC 3.62 Å, phenix 3.66 Å, PDBe FSC server 4.08 Å, and 3DFSC 4.12 Å. This resolution range is reasonable with respect to particle size"

(2) It is not always clear which maps are displayed in each figure (e.g. Figs. 1 & 4), and which maps were used for modeling. For example, in figure 1c, it is stated that the "final cryoSPARC map" was used for modeling. Does this correspond to the non-b-factor-sharpened map ("map") or the b-factor-sharpened map ("map_sharp") output by the Cryosparc non-uniform refinement? From the "Model Building" section of the methods (page 29), it is unclear whether the DeepEMhancer post-processed map was used only for fitting of the ADAT3 N-terminal domain as it seems from the main text. What is the difference between the DeepEMhancer map and the final map referenced in the methods a few sentences later? Please clarify.

For clarification, at all mentions (Legend of Figures 1 and 4 and methods section), we replaced "final cryoSPARC map" with "final non-b-factor-sharpened cryoSPARC map".

Furthermore, we clarified the application of the DeepEMhancer map for fitting of the N-terminal domain (page 17):

Initial: "We jiggle-fitted the ADAT3^N AlphaFold model into the post-processed DeepEMhancer map without further side chain refinement"

Rephrased to: "Since the cryoSPARC map was not sufficient to fit the AlphaFold model, we generated a DeepEMhancer map and jiggle-fitted the ADAT3^N without further side chain refinement."

(3) Because DeepEMhancer processing could in theory cause artifacts that remain undetected when the ground truth is not known (as mentioned in the discussion of Sanchez-Garcia et al. 2021), it would be useful to validate the model fit using another approach. For example, is the ADAT3N fit qualitatively the same when fit into the Cryosparc unsharpened map or a locally filtered version of the map (or another variation of the map in which the ADAT3N region is more visible)?

For biochemical cross-validation of the mode of interaction of ADAT3N with tRNA observed in the model, we conducted RNA electromobility shift assays (EMSA) (Supplementary Figure 4 c-f) and added the **new data** to the manuscript (page 17). We included the newly generated data on these residues into Figure 5b.

Initial: "ADAT3 residues in both these regions have been attributed a role in tRNA binding and deaminase activity in previous experiments (Fig. 5b, Supplementary Data Fig. 4a and Supplementary Data Table 2) (Liu et al., 2020; Ramos-Morales et al., 2021)."

Rephrased to: "Since the maps were not of sufficient quality to unambiguously identify the contact residues, we assessed single point mutants of ADAT3^N of positively charged residues in either of the observed patches for their RNA binding properties. Indeed the ADAT3 mutants K48A, R52E, K164E and R166E weaken the interaction of the ADAT2/3 complex with tRNA in electromobility shift assays (EMSA) (Supplementary Data Figs. 4c and 4d). Additionally, previously reported double and triple mutations of ADAT3 in this region have been attributed a role in tRNA binding and deaminase activity. (Fig. 5b, Supplementary Data Fig. 4a and Supplementary Data Table 2) (Liu et al., 2020; Ramos-Morales et al., 2021)"

Furthermore, we altered the view window of Supplementary Data Figure 4b, comparing the sharpened and non-sharpened maps, to better illustrate the fit of the alpha fold model of ADAT3^N based on restraints from the linker regions.

(4) Regarding the "RY-gate" (page 12), it's not clear that one can interpret the "gating" activity to be important. An alternative explanation, consistent with the activity assays presented here, would be that the positive charge at position 159 is important for some other reason, such as interacting with the tRNA substrate as or after it enters the active site. The authors should address this in the text.

We interpret the gating activity - not only the charge - as important based on the following evidence: first, if R159 would be directly involved in tRNA binding, we would expect to see a strong density for its side chain in this region of our EM map, which we don't; supporting our observation, the Afonzo Lab showed in a previous paper that the mutant R159A has an unaltered tRNA binding affinity (Ragone et al., 2011). Second, if only the charge would be important, we would expect to see equal activity of the wt and the charge-conserving R159K mutant. The only difference between an Arginine and a Lysine in this position is its ability to form a cation- π interaction with Y205, as observed in the available substrate-free structures. The only plausible explanation for "less active" eukaryotic residues compared to the bacterial sequence is a gating activity, restricting the active site access. For clarification we added additional evidence from the earlier study (Ragone et al. 2011) and rephrased two paragraphs:

Initial: "To test the importance of the gate residues, we measured TbADAT2/3 deamination activity of the wild type enzyme and several mutants."

Rephrased: "In an earlier study, the substitution of R159^{TbADAT2} to alanine displayed unaltered RNA binding properties (Ragone et al., 2011), so next we set out to scrutinize the importance of the gate residues in catalysis through deamination assays." (pages 12 and 13)

Initial: "A mutant mimicking the bacterial situation with a R159K substitution, strikingly, showed an almost 3-fold improved reaction rate compared to the wild type, suggesting a lysine in this position is more efficient for deamination activity (Figs. 3c, 3d, Supplementary Data Figs. 3j and 3k)."

Rephrased to: "A mutant mimicking the bacterial situation with a R159K substitution, conserving the positive charge but altering its geometry, strikingly, showed a more than 2-fold improved reaction rate compared to the wild type, suggesting a lysine in this position is more efficient for deamination activity (Figs. 3c, 3d, Supplementary Data Figs. 3j and 3k). (page 13)

(5) How much adjustment of the AlphaFold model was necessary to fit the KR motif into the EM density? From Fig. 4 and the validation report, the fit of these residues is on the poorer side - rigid body fitting would seem reasonable there. Additionally, from the view shown in Fig. 4a, it isn't clear whether the density extending out from the tRNA (marked with the blue arrow) is really due to the KR loop, given the lack of continuous density for the side chains. Is there another threshold that can be displayed (in this figure or the supplement) to make this convincing?

A weakness of the conformation of the computational model is that it does not take the complex formation with RNA into account. We used the orientation of side chain densities extruding from the main chain density to correctly orient the main chain of the model. To illustrate the minor manual adjustments made to the alpha fold model in ADAT2^C, we **added Supplementary Data Figure 1g**.

To avoid misinterpretation, we intentionally chose to display a backbone-only representation of ADAT2^C. The essential role of the KR-motif for tRNA interaction has been confirmed in an earlier study (Ragone et al. 2011).

For consistency and to avoid misinterpretation, we display all maps in the paper figures at the same threshold. We **removed the blue arrows** from Figure 4 to avoid misinterpretation.

For completeness, we calculated rmsd values for all domains of the protein complex as measure for the model adjustments made during the refinement and **added it as a panel in Supplementary Data Figure 1h**.

(6) In the “Cryo-EM Data Processing” section of the methods (page 28), the authors state “One round of 3D classification generated three ab-initio reconstructions”. Could the authors please rephrase this to clarify whether they used the “ab initio reconstruction” job within Cryosparc or the “3D classification” job?

We rephrased to clarify (page 29):

Initial: “One round of 3D classification generated three ab-initio reconstructions followed by hetero refinement, from which the main class, with 427,165 particles were 2D classified. Blurred 2D classes were discarded and the remaining 379,795 particles were once more 3D classified by generating 4 ab-initio reconstructions followed by hetero refinement.”

Rephrased to: "One round of 3D classification generated three classes using ab-initio reconstruction, followed by hetero refinement. The 427,165 particles from the main class were 2D classified and blurred 2D classes were discarded. The remaining 379,795 particles were once more 3D classified by generating 4 ab-initio reconstructions followed by hetero refinement."

Minor comments:

(1) Regarding the discussion of inhibition by non-substrate tRNAs on p. 20: This is an interesting aspect of this biology. Is anything known about levels of ADAT2/3 or the various tRNAs in the cytoplasm?

The database (<https://www.yeastgenome.org/>) indicates 1336 and 1205 molecules per yeast cell for ADAT2 and ADAT3, respectively, yielding a concentration of around 100 nM. Unfortunately, not much is known about the steady-state levels of ADAT2/3 in trypanosoma cells, but if one goes by the convention that in cells many soluble enzymes are at concentrations that approach their K_m , then one would predict that ADAT2/3 is at the high nM level. This implies that the total tRNA population (in the high μ M range) far exceeds that of the enzyme. As far as the tRNAs are concerned, Tan et al., Mol. Cell. Biol. 2002, quantified the levels of different tRNAs in trypanosomes representing 15 different amino acids. Out of those only one was an A₃₄-containing tRNA (tRNA^{Leu}AAG), which is a substrate for ADAT2/3. Fortunately, they determined the levels for all 4 tRNA^{Leu} isoacceptors (anticodons AAG, CAA, CAG and UAG). They found the following distribution: AAG-130K molecules per cell, CAA-220K, CAG-121K, and UAG-60K. Thus, the non-A₃₄-containing tRNA^{Leu} are at 4X higher concentration. If we also consider the fact that only 8 out of 60 possible tRNAs have A₃₄ and given that *in vitro* there is little difference between binding a substrate versus a non-substrate, then intracellular competition, or lack thereof, by ADAT2/3 versus other tRNA interacting factors is of biological significance but will remain a question for future studies.

(2) Why weren't the insect cell expressed ADAR2/3 proteins used for the biochemical assays? Do the solubility-increasing point mutations affect activity, or is there another reason?

In the Kowalinski lab, protocols for high-yield insect cell expression have been established and optimized, initially to produce sufficient amounts for crystallization trials. For practical reasons we later kept using these protocols also for cryo-EM grid preparation. The Alfonzo laboratory is not equipped with a facility to express protein complexes in insect cells, so all the mutants used for kinetics in this work were generated and expressed in *E. coli* instead. All expression constructs (insect and *E. coli*) carry the A87G/C117S mutation, which prevents homodimerization of ADAT2. The variant used is identical in kinetic behavior to the wild-type enzyme and has been used in numerous papers (>10 years) from the Alfonzo lab for RNA binding and kinetic analysis.

We clarify in the methods section:

Initial: "For structure determination full-length ADAT2 (*Tb927.8.4180*, with two point mutation that improve solubility: A87G/C117S) and full-length His-tagged ADAT3 (*Tb927.11.15280*) [...]".

Rephrased to (page 25): "For structure determination full-length ADAT2 (*Tb927.8.4180*, with two point mutations that improve solubility and reduced ADAT2 homodimerization: A87G/C117S) and full-length His-tagged ADAT3 (*Tb927.11.15280*) [...]".

Initial: "Briefly, the coding sequences of *TbADAT2* and *TbADAT3* were cloned into expression vectors pETDuet-1 and pET-28(a)+ and used to transform *E. coli* BL21-IRL strain."

Rephrased to (page 26): "Briefly, the coding sequences of *TbADAT2* (A87G/C117S) and *TbADAT3* were cloned into expression vectors pETDuet-1 and pET-28(a)+ and used to transform *E. coli* BL21-IRL strain."

(3) In the "Cryo-EM Data acquisition" section of the methods (p. 28): the "dose" should be more precisely referred to as "electron exposure rate" and "total dose" should be more precisely referred to as "total electron exposure".

The text was corrected (page 28):

Initial: "Movies were acquired for 10 seconds at a dose of 3 e/pixel/s, resulting in a total dose of 55.00 e/A² at the sample level, fractionated into 80 movie frames."

Rephrased to: "Movies were acquired for 10 seconds at an electron exposure rate of 3 e/pixel/s, resulting in a total electron exposure of 55.00 e/A² at the sample level, fractionated into 80 movie frames."

Reviewer #2 (Remarks to the Author):

Here Kowalinski and colleagues report the cryoEM structure of the *T. brucei* tRNA-specific adenosine deaminase ADAR2/3 heterodimer bound to a model substrate tRNA. ADAR2/3 selectively deaminates A34 in the anticodon loops for several eukaryotic tRNAs and is related to the bacterial TadA enzyme that carries out a similar reaction at A34 of tRNA(Arg) as a homodimer. While structures of bacterial TadA, free and bound to RNA, have been known for many years and its mechanism for substrate recognition is well established, how the eukaryotic ADAR2/3 heterodimer reacts with a diverse group of tRNAs that vary in their anticodon loop sequence was not known (prior to the work reported here). This structure and accompanying biochemical data constitute an important advance in our understanding of substrate recognition by RNA modifying enzymes in general and specifically how eukaryotes modify the A34 position of several different tRNAs. This is a well written paper. The structure both explains data already reported in the literature and generates new hypotheses for the mechanism of ADAR2/3 substrate recognition and processing. I have only the following one critical comment.

(1) A key distinction between the bacterial system and the eukaryotic system described here is the observation of contacts between the ADAR3 component of the heterodimer and the D arm and T arm of the tRNA. The authors state that others have identified ADAR3 residues important for RNA binding and present a table summarizing those results (Extended data table 2) and a figure showing the locations of these residues (Fig. 5b). However, it appears their structure suggests that some of these residues may be involved in direct contact with RNA while others may not. More discussion is needed about what their structure says about what specific residues are likely involved in direct RNA contacts in these locations. If their data are insufficient to identify specific amino acid-nucleotide contacts in these regions, this should also be stated.

To identify more precisely single contact amino acid residues and cross validate the model, we tested single point mutants of the complex via electromobility shift assays (EMSA), and **added the data of this new experiment in Supplementary Data Figures 4c and 4d**. For clarification, we have rephrased the manuscript (page 17), and also indicate the addressed residues in Figure 5b.

Initial: "ADAT3 residues in both these regions have been attributed a role in tRNA binding and deaminase activity in previous experiments (Fig. 5b, Supplementary Data Fig. 4a and Extended Data Table 2) (Liu et al., 2020; Ramos-Morales et al., 2021)."

Rephrased to: "Since the maps were not of sufficient quality to unambiguously identify the contact residues, we assessed single point mutants of ADAT3^N of positively charged residues in either of the observed patches for their RNA binding properties. Indeed the ADAT3 mutants K48A, R52E, K164E and R166E weaken the interaction of the ADAT2/3 complex with tRNA in electromobility shift assays (EMSA) (Supplementary Data Figs. 4c and 4d). Additionally, previously reported double and triple mutations of ADAT3 in this region have been attributed a role in tRNA binding and deaminase activity. (Fig. 5b, Supplementary Data Fig. 4a and Supplementary Data Table 2) (Liu et al., 2020; Ramos-Morales et al., 2021)."

Reviewer #3 (Remarks to the Author):

In this manuscript, Dolce et al. have determined the structure of ADAT2/3 in complex with its full-length tRNA substrate revealing insight into the mechanisms of tRNA A-to-I editing in eukaryotes. Thereby, they visualize how the N-terminal domain of ADAT3 helps to recognize the entire tRNA forming interactions in particular with the D arm. Moreover, they identify a novel gate of two amino acid residues in the active site of ADAT2 that is closed in the RNA-free structure (published previously) but opens to accommodate the anticodon arm in the active site. The importance of these amino acid residues is supported by biochemical studies.

In general, the experiments are technically sound. The manuscript is overall well written and compares the new structure to previously reported structures of apo ADAT2/ADAT3 and the bacterial homolog TadA bound to an anticodon stem-loop. While this manuscript provides new insight into the molecular details how ADAT2/3 recognize tRNA, the manuscript in its current format is could be further improved to broaden the scope and impact. In particular, several questions around ADAT2/3 are not clearly addressed by the authors as outlined below.

1. The authors should describe more clearly the current understanding how ADAT2/3 discriminate cognate from non-cognate tRNA. This is in particular important for readers who are generally interested in RNA recognition and modification, but who are not experts on A-to-I editing. Based on my own reading (e.g. HR Frigole et al, RNA 2019), it seems that ADAT2/3 recognizes and modifies all eukaryotic tRNA which have an A at position 34. Thus, it might not be required for ADAT2/3 to recognize any other structure- or sequence-elements in tRNA except for ensuring that a tRNA is bound rather than any other RNA with an anticodon-like loop. This is eluted to in the

manuscript but can be stated much more clearly to set the stage for the discussion of tRNA recognition.

The reviewer is correct in that indeed ADAT2/3 can deaminate tRNAs with A₃₄. Unfortunately, currently it is not clear how the enzyme discriminates cognate from non-cognate substrates. In passing, the same is true with many tRNA modifying enzymes. The Alfonzo lab has shown through kinetic assays that G₃₄ containing tRNA can bind with the ~same affinity as A₃₄ containing tRNAs, suggesting non-substrate tRNAs can bind efficiently (Ragone et al., 2011 RNA). Ramos-Morales et al., NAR 2021, shows that ADAT2/3 binds to diverse cognate and non-cognate tRNA with similar affinity. But just to be clear, ADAT2/3 cannot deaminate shorter versions of a tRNA and if one were to make mutations of the various tRNA domains, it fails to bind efficiently (Auxilien and Grosjean J Mol Biol. 1996). So, it does require a properly folded (L-shape) full length molecule and recognizes its overall structure, which will be material for follow-up studies.

We add a sentence in the introduction to clearly state that ADAT2/3 can bind all tRNAs:

Added (page 5): All tRNAs meeting these requirements, target and non-target tRNAs, can interact with ADAT2/3, yet, in a previous study, an A₃₄ containing substrate displayed a ten times faster dissociation compared to other nucleotides (Ragone et al., 2011).

Furthermore, we clarified two sentences in the discussion:

Initial (page 20): “Validated tRNAs can then enter through the molecular RY-gate,[...]”.

Rephrased to: “Once validated as any tRNA carrying the correct geometry, the substrate can then enter through the molecular RY-gate,[...]”.

Initial (page 21): “With the inserted anticodon loop, the unstructured portions in the ADAT2 C-terminus, notably the positively charged and essential KR-motif, adopt a folded conformation and help to correctly place the tRNA in a manner that is amenable for catalysis (Fig. 6c). For cognate tRNAs, the L-shape architecture, anticodon loop geometry, and the A₃₄ nucleotide will perfectly align to the three main binding motifs in ADAT2/3, namely ADAT3^N, ADAT2^C and joint deaminase core, to trigger the deamination reaction.”

Rephrased to: “With the inserted anticodon loop, the unstructured portions in the ADAT2 C-terminus, notably the positively charged and essential KR-motif, adopt a folded conformation and help to correctly place the tRNA in a manner that is amenable for catalysis (Fig. 6c). At this point, non-cognate tRNAs (i.e. all tRNAs with G, U or C in position 34) are eventually released, but for cognate tRNAs, the L-shape architecture, anticodon loop geometry, and the A₃₄ nucleotide will perfectly align to the

three main binding motifs in ADAT2/3, namely ADAT3^N, ADAT2^C and joint deaminase core, to trigger the deamination reaction.”

2. Why did the authors choose a non-cognate tRNA^{Thr} for the structural studies rather than transcribing a cognate tRNA with an A34 to C mutation? It cannot be ruled out that such a cognate tRNA could be positioned slightly differently on ADAT2/3.

In our hands, the *in vitro* transcribed non-cognate tRNA^{Thr} was stable, correctly folded and formed stable complexes suitable for structure determination. As stated in the manuscript (page 23), we indeed expect different tRNAs to interact slightly differently: “[...] the distal eukaryote-acquired RNA binding features show a certain flexibility either via connection through linkers (ADAT3^N) or induced-fit adaptation of the protein portion (ADAT2^C). This flexibility is most likely the key to the accommodation of multiple tRNAs with divergent anticodon loop sequences.” Additionally, Ramos-Morales et al., NAR 2021, show that ADAT2/3 is able to deaminate a non-cognate tRNA^{Gly}_{CCC}, with the mutation C34A. Similar studies by Auxilien and Grosjean (J Mol Biol. 1996) in yeast showed that any of the isoacceptors with nucleotides other than A34 but belonging to the same tRNA type when mutated to contain A34 become substrates for deamination.

3. The structure provides limited insight into the active site and catalysis since the catalytic glutamate is disordered and a C34 is used to prevent catalysis. Is it possible to generate a structure using a cognate tRNA substrate with an A34, but mutating the catalytic glutamate in ADAT2? Or in other words, can their structure provide insight into catalysis? Or has this been previously deciphered based on the structure of bacterial TadA with RNA bound?

With the given resolution of our map we concentrate our study on global tRNA binding features of ADAT2/3, which had not yet been deciphered. The dissection of enzyme catalysis would require a higher, atomic resolution structure to be able to interpret side chain orientation and bond configuration. Moreover, the mechanism of hydrolytic deamination has been proposed earlier together with the crystal structure of TadA bound to an RNA hairpin for TadA. In that study, the target base had been replaced by the non-hydrolyzable adenosine analog nebularine to create a stable complex (Loosey et al., 2006). Taking into account the conservation of the active site between TadA and ADAT2/3 we would expect a conserved catalytic mechanism (Figure 2 e, f, g, Supplementary Figure 3c).

Using a substrate containing the proper A₃₄ in combination with a catalytically inactive mutant of ADAT2 (E92A), as suggested, would be a potential alternative approach (see Ragone et al., 2011) to yield a stable trimeric complex. However, due to the same technical limitations related to the size and asymmetry of the particle, we doubt that this approach would help to improve the resolution sufficiently to provide insight

into catalysis. To clarify about the conservation between ADAT2/3 and TadA, we rephrased in the main text:

Initial: “The active site of *Tb*ADAT2 features the typical CDA elements: a Zn²⁺ cation coordinated by two cysteines (C136^{*Tb*ADAT2} and C139^{*Tb*ADAT2}) and a histidine (H90^{*Tb*ADAT2}).”

Rephrased to (page 11): “The active site of *Tb*ADAT2 features the typical CDA elements: a Zn²⁺ cation coordinated by two cysteines (C136^{*Tb*ADAT2} and C139^{*Tb*ADAT2}) and a histidine (H90^{*Tb*ADAT2}), and the catalytic glutamate (E92^{*Tb*ADAT2}), suggesting a similar catalytic mechanism as described for TadA (Losey et al., 2006).”

4. Interestingly, the authors claim that a key difference between bacterial TadA and eukaryotic ADAT2/3 lies in the recognition of the anticodon loop which is sequence-specific for TadA, but sequence-independent for ADAT2/3. To corroborate this claim, the authors should include and discuss additional figures highlighting the changes in ADAT2/3 compared to TadA that allow the recognition of each anticodon base without conferring sequence specificity.

Supplementary figures 5a and 5b compare TADA and ADAT2/3 residues that interact with the tRNA respecting the resolution limits of the model. Generating more detailed illustrations explicitly featuring side chain contacts to the RNA would potentially lead to over-interpretation of the model at the given resolution by large parts of the audience. To make our case on the sequence-independent recognition clear, we rephrase a paragraph of the discussion on page 22 and 23.

Initial: “It has been proposed that substrate diversification to multiple tRNAs by ADAT2/3 was the result of active site mutations that effectively relaxed the need for sequence-specific recognition (Elias and Huang, 2005). A feasible evolutionary scenario for increased substrate diversity by the eukaryotic enzymes may have involved a gene-duplication of the ancestral ADAT2 protein to yield ADAT3 and the enzyme acquired mutations that lead to active-site relaxation (Auxilien et al., 1996; Elias and Huang, 2005; Gerber and Keller, 1999; Rubio et al., 2007; Torres et al., 2021; Zhou et al., 2014). In this intermediate state, ADAT2 was still likely able to catalyze the reaction in a single substrate *à la* TadA. However, appearance of the ADAT3 subunit, and its ability to heterodimerize, allowed it to abandon sequence-specific recognition and acquire additional sequence-independent RNA binding elements located distant from the catalytic center, reducing the importance of the active-site in tRNA recognition and finally leading to a multi-step binding mechanism. We observe that while the bacterial TadA homodimer almost exclusively interacts with the anticodon loop bases C₃₂-A₃₈, the additional eukaryotic domains add manifold sequence-nonspecific contacts along the anticodon arm up to the elbow region of the tRNA (Fig. 5C and Supplementary Data Fig. 5).”

Rephrased to: "It has been proposed that substrate diversification to multiple tRNAs by ADAT2/3 was the result of a gene-duplication of the ancestral ADAT2 protein yielding ADAT3. The heterodimer could now acquire active site mutations that effectively relaxed the need for sequence-specific recognition and would allow for the acquisition of tRNA binding domains distal from the active site (i.e. ADAT2^C and ADAT3^N). (Auxilien et al., 1996; Elias and Huang, 2005; Gerber and Keller, 1999; Rubio et al., 2007; Torres et al., 2021; Zhou et al., 2014). However, in the tRNA bound structure we do not observe direct evidence for a relaxation of the active site, but a rather high conservation to the ancestral enzyme. A rather rigid central pocket for the A₃₄ is observed when comparing it to the free enzyme structures. Furthermore, the Supplementary anticodon binding pocket of both, the bacterial TadA homodimer and the ADAT2/3 heterodimer, display similar ways of interaction with the splayed out anticodon loop bases C₃₂-A₃₈, with no explicit differences explaining their different substrate preferences. However, as a major difference, the eukaryotic enzyme has acquired additional domains contributing multiple additional sequence-nonspecific interactions along the anticodon arm up to the elbow region of the tRNA (Fig. 5C and Supplementary Data Fig. 5). The appearance of these new interaction motifs potentially lead to a multi-step binding mechanism and may have reduced the dominance of the active site for tRNA binding. Thus, jointly all RNA binding elements in the heterodimer may even provide sufficient affinity to guide substrates with imperfect sequences into the active site, as long as they satisfy the need for an overall tRNA geometry.

5. When investigating the importance of the RY-gate, the authors make several single-residue substitutions. Do these substitutions affect protein stability and folding? The authors should assess this by melting curves or CD spectroscopy to show that the ADAT2/3 variants are correctly folded.

The **CD data has been generated on request and has been added in Supplementary Data Figure 3**. No significant differences in the spectra of the mutants were observed.

6. Have the corresponding substitutions of the RY gate been made in bacterial TadA and what are their effects?

A role in RNA binding has been previously attributed to the positively charged bacterial "gate" residue without experimental validation (Losey et al. 2006). For the aromatic residue, several studies exist that used random mutation evolution screens to generate functional synthetic TadA variants for genome editing. Here, amongst others, the F-to-Y of the aromatic residue corresponding to the "gate" position has been identified (TadA8e octuple mutant, Richter et al., Nature Biotechnology, 2020). While this is an interesting exemplification, adding this information to our study might cause more confusion than insight. The gate seems an acquired feature which is conserved

to eukaryotic ADATs, discovered through our gain-of-function mutant when we reverted the residues to the bacterial sequence. In the future, the vice-versa experiment, introducing the eukaryotic “gate” into the bacterial enzyme, will be an interesting follow up of study.

7. Figure 3 c and d are redundant as the same information (krel as a representation of kobs) is shown twice.

The kobs table was moved to the supplement. Mutants without any detectable activity were added to the plot, as we consider their inactivity relevant.

8. To determine kobs, the authors must measure time courses where more than one data point is outside the endlevel (Extended Data Fig 3). In other words, there should be more data points in the first 10 min.

The kinetics have **been repeated and more data points have been added**

9. E. Ramos-Morales et al., NAR 2021, reported the structure of the ADAT2/ADAT3 complex in absence of RNA and predicted the interaction with tRNA (graphical abstract and Fig. 5). In this manuscript, the authors should discuss how their cryo-EM structure differs or confirms this previously published model which looks very similar in general. Here, the authors state that “all previously observed positions (of the ADAT3 N terminal domain?) disagree with our cryo-EM map”. This statement should be supported by a comparative figure.

To make our statement clear, we **add Supplementary data figure 6h** as requested for comparison and rephrase in the main text (page 16):

Initial: "However, all previously observed positions disagree with our cryo-EM map or would clash with other parts of the model."

Rephrased to: "However, all previously experimentally observed positions of ADAT3^N disagree with our cryo-EM map or would clash with other parts of the model (Supplementary Data Fig. 6h)."

Furthermore, **we conducted new RNA electromobility shift assays**, confirming the interactions identified from our model (Supplementary Figure 4 c-f). In comparison to the previous studies using double or triple mutations of ADAT3, we could precisely identify single residues with a role in tRNA binding (K48A, R52E, K164E and R166E).

10. In the introduction, the authors mention mutations in ADAT3 that cause a rare disorder. What insight does the ADAT2/3-tRNA structure provide regarding the impact of these mutations on ADAT2/3 structure and function? The authors should map these mutations onto the structure and discuss possible effects.

The direct effect of the human disorder mutation ADAT3 V128M on RNA binding has been addressed earlier by Ramos-Morales et al., NAR 2021.

We added the following sentence to the manuscript (page 18): "The human disease mutation V128M maps to ADAT3^N (V139 in *TbADAT3*) and seems not in direct contact with the tRNA. This observation agrees with an earlier study showing that the mutant retained its tRNA binding properties, but showed subtly diminished levels of deamination most probably due to a slightly altered overall geometry (Ramos-Morales et al., 2021)."

11. This structure unambiguously shows that ADAT2/3 can tightly bind to non-cognate tRNA. In the discussion, the authors speculate why this may not lead to competitive inhibition in the cell, but their argumentation is not convincing. As much as non-cognate tRNAs may be "sequestered from the available cytosolic pool by other tRNA targeting enzymes", the same will hold true for cognate tRNAs. Thus, this structure raises almost more questions than answers regarding the discrimination of cognate and non-cognate tRNAs by ADAT2/3. To address this issue, the authors may conduct competition experiments in the presence of both cognate and non-cognate tRNAs to at least quantify whether competition takes place.

We agree that this is indeed an aspect to follow-up. Most importantly, *in vivo* studies have to be addressed, as these will also account for the "real" availability of tRNAs in the cellular environment, as despite their high copy number, they might mostly not exist in free form but rather in complex with tRNA binding proteins and cellular machineries. We rephrased our discussion to clarify that the idea is purely speculative and rather material for future more detailed kinetic studies with different chimeric tRNAs (page 21):

Initial: "Here, we can only assume that the effective concentration of free A₃₄-tRNAs makes the difference"

Rephrased to: "Here, we can only speculate that the effective concentration of free A₃₄-tRNAs could possibly make the difference"

REVIEWERS' COMMENTS

Reviewer #1 (Remarks to the Author):

The authors have adequately addressed my comments. Additionally, I thank the authors for replying properly to Minor Comment #2 despite my ADAR/ADAT typo. I recommend the revised manuscript for publication.

Reviewer #3 (Remarks to the Author):

The authors are presenting a significantly revised manuscript that takes most considerations of the reviewers into account and addresses these through new experiments and revised writing.

In particular, the interpretation of the interaction of the N-terminus of ADAT3 with tRNA is greatly improved by the addition of electrophoretic mobility shift assays (EMSA) with single-residue substitutions in ADAT3.

Similarly, I appreciate that the authors validated the protein variant's fold by CD spectroscopy, and that they repeated the time courses of deamination with more time points.

The revisions to the text and the figures improve the manuscript and clarify many questions by all three reviewers.

For all these reasons, I recommend publishing this manuscript.

Having said this, I continue to have questions about the 'specificity' of ADAT2/3, but these don't necessarily have to be addressed in the manuscript.

Based on the data, it seems that ADAT2/3 does not discriminate at all between tRNAs with respect to binding, and that it only discriminates against tRNAs lacking the A34 with respect to catalytic activity.

Thus, the main reason for recognizing the overall tRNA shape is likely to discriminate against the editing of OTHER cellular RNAs that may have a 7-residue loop with an A in the position corresponding to A34.

The authors may or may not wish to clarify this in the manuscript.

Dear reviewers,

All authors would like to express their gratitude to you for your precious time and energy taken to help to improve our manuscript further.

Thank you,
Eva Kowalinski (in the name of all authors)

In the following we address all reviewers' comments:

Reviewer #1 (Remarks to the Author):

The authors have adequately addressed my comments. Additionally, I thank the authors for replying properly to Minor Comment #2 despite my ADAR/ADAT typo. I recommend the revised manuscript for publication.

Thank you for your time to revise our manuscript!

Reviewer #3 (Remarks to the Author):

The authors are presenting a significantly revised manuscript that takes most considerations of the reviewers into account and addresses these through new experiments and revised writing.

In particular, the interpretation of the interaction of the N-terminus of ADAT3 with tRNA is greatly improved by the addition of electrophoretic mobility shift assays (EMSA) with single-residue substitutions in ADAT3.

Similarly, I appreciate that the authors validated the protein variant's fold by CD spectroscopy, and that they repeated the time courses of deamination with more time points.

The revisions to the text and the figures improve the manuscript and clarify many questions by all three reviewers.

For all these reasons, I recommend publishing this manuscript.

Having said this, I continue to have questions about the 'specificity' of ADAT2/3, but these don't necessarily have to be addressed in the manuscript.

Based on the data, it seems that ADAT2/3 does not discriminate at all between tRNAs with respect to binding, and that it only discriminates against tRNAs lacking the A34 with respect to catalytic activity.

Thus, the main reason for recognizing the overall tRNA shape is likely to discriminate against the editing of OTHER cellular RNAs that may have a 7-residue loop with an A in the position corresponding to A34.

The authors may or may not wish to clarify this in the manuscript.

Dear reviewer, this is exactly the question that we are also currently following up in further experiments! We add a sentence in the discussion:

Initial: Once validated as any tRNA carrying the correct geometry, the substrate can then enter through the molecular RY-gate, formed by cation- π interactions between the gate residue side chains, into the anticodon binding cleft of the joint ADAT2/3 deaminase core.

Rephrased: Once validated as any tRNA carrying the correct geometry, the substrate can then enter through the molecular RY-gate, formed by cation- π interactions between the gate residue side chains, into the anticodon binding cleft of the joint ADAT2/3 deaminase core. Supposedly, other cellular, non-tRNA 7-nucleotide RNA hairpins, resembling an anticodon loop, would not be able to overcome the gate in lack of the necessary contacts to ADAT3N.